# MDN: Parallelizing Stepwise Momentum for Delta Linear Attention

**Yulong Huang** [* 1]  **Xiang Liu** [* 1]  **Hongxiang Huang** [1]  **Xiaopeng Lin** [1]  **Zunchang Liu** [1]
**Xiaowen Chu** [1]  **Zeke Xie** [✉ 1]  **Bojun Cheng** [✉ 1]

## Abstract

Linear Attention (LA) offers a promising paradigm for scaling large language models (LLMs) to long sequences by avoiding the quadratic complexity of self-attention. Recent LA models such as Mamba2 and GDN interpret linear recurrences as closed-form online stochastic gradient descent (SGD), but naive SGD updates suffer from rapid information decay and suboptimal convergence in optimization. While momentum-based optimizers provide a natural remedy, they pose challenges in simultaneously achieving training efficiency and effectiveness. To address this, we develop a chunkwise parallel algorithm for LA with a stepwise momentum rule by geometrically reordering the update coefficients. Further, from a dynamical systems perspective, we analyze the momentum-based recurrence as a second-order system that introduces complex conjugate eigenvalues. This analysis guides the design of stable gating constraints. The resulting model, Momentum DeltaNet (MDN), leverages Triton kernels to achieve comparable training throughput with competitive linear models such as Mamba2 and KDA. Extensive experiments on the 400M and 1.3B parameter models demonstrate consistent performance improvements over strong baselines, including Transformers, Mamba2 and GDN, across diverse downstream evaluation benchmarks. Code: [github.com/HuuYuLong/MomentumDeltaNet](github.com/HuuYuLong/MomentumDeltaNet).

---

[*]Equal contribution  [1]The Hong Kong University of Science and Technology (Guangzhou), Guangzhou, China. Email: yhuang496@connect.hkust-gz.edu.cn,  xliu886@connect.hkust-gz.edu.cn.  Correspondence to: Zeke Xie <zekexie@hkust-gz.edu.cn>, Bojun Cheng <bocheng@hkust-gz.edu.cn>.

*Proceedings of the 43$^{rd}$ International Conference on Machine Learning*, Seoul, South Korea. PMLR 306, 2026. Copyright 2026 by the author(s).

## 1. Introduction

The Transformer architecture has become the cornerstone of modern deep learning, owing to the inherent parallelizability of training for sequence modeling (Vaswani et al., 2017). However, the self-attention layers within the Transformer suffer from the quadratic scaling ($O(L^2)$) with respect to the sequence length ($L$) (LI et al., 2025), severely limiting scalability in long context scenarios (Hsieh et al., 2024). To overcome this limitation, Linear Attention (LA) has emerged as a promising paradigm by reformulating the Softmax operator into linear kernel functions (Schlag et al., 2021), reducing complexity to $O(L)$ time and maintaining constant sized inference states. Although early LA suffered from limited expressive power, recent recurrent update mechanisms, notably the Decay Rule (e.g., Mamba (Dao & Gu, 2024), GLA (Yang et al., 2024a)) and the Delta Rule (e.g., GDN (Yang et al., 2025), KDA (Team et al., 2025)) have substantially narrowed the performance gap relative to Transformers. Coupled with hardware efficient chunkwise parallelism, these advancements have enabled the development of hybrid large language models (LLMs) that deliver superior throughput while maintaining competitive effectiveness (Lieber et al., 2024; Gu et al., 2025; Team, 2025; Wang et al., 2025a; Bae et al., 2025; Liu et al., 2026b).

However, current LA mechanisms still struggle to capture fine-grained historical details (Wen et al., 2024), as reflected in their limited capability in context retrieval tasks (Allen-Zhu, 2025). From the Test-Time Training (TTT) perspective (Sun et al., 2020; 2024), the recurrence formulation of LA can be interpreted as a closed-form solution for the online optimization of a latent objective (Wang et al., 2025b). Specifically, mechanisms such as the Decay and Delta rules correspond to latent loss objectives with weight decay and $L_2$ MSE loss, respectively (Zhong et al., 2025). However, their updates are invariably derived via naive Stochastic Gradient Descent (SGD). Therefore, the retrieval limitations of existing LA models can be partially attributed to the inherent constraints of this oversimplified SGD update mechanism.

The advantages of momentum-based optimizers over naive SGD are well established in the optimization literature (Nesterov, 1983; Kingma, 2014; Liu et al., 2025). While SGD relies solely on instantaneous gradients and is therefore sen-

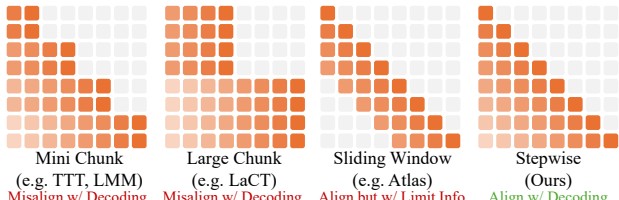

| Mini Chunk | Large Chunk | Sliding Window | Stepwise |
| :---: | :---: | :---: | :---: |
| (e.g. TTT, LMM) | (e.g. LaCT) | (e.g. Atlas) | (Ours) |
| Misalign w/ Decoding | Misalign w/ Decoding | Align but w/ Limit Info. | Align w/ Decoding |

*Figure 1.* Comparison of causal structures during training across different momentum update schemes. Blockwise scheme (e.g, TTT (Sun et al., 2024), LMM (Behrouz et al., 2025b) and LaCT (Zhang et al., 2025)) introduces intra-block non-causality, causing a training-inference mismatch. Sliding window scheme like Altas (Behrouz et al., 2025a) limits the truncated historical context. Our Stepwise Momentum maintains a strict causal mask, ensuring exact consistency between parallel training and decoding.

sitive to gradient noise (Sclocchi et al., 2023), momentum methods (Polyak, 1987) accumulate gradient information in an auxiliary hidden state, which can attenuate noise, smooth updates, and stabilize the optimization trajectory (Sutskever et al., 2013). Since the recurrence in linear attention admits an online optimization interpretation (Liu et al., 2024), accumulated gradients in momentum provide access to longer historical information. From this perspective, incorporating momentum offers a potential direction for improving representation robustness and retrieval performance.

While momentum is straightforward to implement in recurrent form, efficiently parallelizing it for large-scale training remains challenging. Prior non-linear RNNs typically resort to blockwise momentum updates to improve hardware utilization (Figure 1), sacrificing strict temporal causality for throughput. Increasing the block size weakens intra-block dependency modeling, leading to degraded performance due to training–inference mismatch. In contrast, stepwise momentum (block size of 1) preserves causality and yields the strongest empirical performance (Sun et al., 2024), but its sequential updates make it impractical for large scale pretraining. This tension between causality and parallel efficiency motivates the need for a scalable parallelization strategy that retains the benefits of stepwise momentum.

In this work, we propose a chunkwise parallel algorithm for the stepwise momentum rule. The algorithm decouples recursive update coefficients from a geometrical perspective and enables efficient parallel computation while preserving strict causality. We further formulate the momentum rule as a second order dynamical system, revealing that momentum introduces complex eigenvalues into the recurrence dynamics and guiding the design of constrained gating mechanisms. Finally, by combining the efficient chunkwise parallel algorithm with the proposed effective gating constraints, we introduce Momentum DeltaNet (MDN). The Triton-based implementation achieves training efficiency comparable to competitive linear models such as KDA and Mamba2. Experiments at the 400M and 1.3B scales show consistent performance gains over various strong baselines.

## 2. Notation and Preliminaries

We use bold upper-case letters $(\mathbf{Q}, \mathbf{K})$ for matrices and bold lower-case letters $(\boldsymbol{q}, \boldsymbol{k})$ for column vectors. A sequence of length $L$ is divided into $L/C$ chunks of size $C$. State matrices are re-indexed such that $\mathbf{S}_{[t]}^i = \mathbf{S}_{tC+i}$ for $t \in [0, L/C]$ and $i \in [1, C]$. For convenience, we denote $\mathbf{S}_{[t]} := \mathbf{S}_{[t]}^0 = \mathbf{S}_{[t-1]}^C$, signifying that the initial state of the current chunk is equivalent to the final state of the preceding chunk.

For any scalar sequence $\{x_k\}$, the global and intra-chunk cumulative products are defined as $\bar{x}_k := \prod_{j=1}^k x_j$ and $\bar{x}_{[t]}^r := \prod_{j=1}^r x_{[t]}^j$, respectively. We define the chunk-level vector $\bar{\boldsymbol{\alpha}}_{[t]} := [\bar{\alpha}_{[t]}^1, \dots, \bar{\alpha}_{[t]}^C]^\top \in \mathbb{R}^C$, and use $\bar{\boldsymbol{\alpha}}_{[t]}^{i \to j}$ to denote the sub-vector covering indices $1 \leq i < j \leq C$. Further details regarding the notation are provided in § A.

### 2.1. From Self-Attention to Linear Attention

**The Self-Attention** mechanism enables the autoregressive Transformers to capture temporal dependencies (Vaswani et al., 2017). For an input sequence $\mathbf{X} \in \mathbb{R}^{L \times d_1}$, the output $\mathbf{O} \in \mathbb{R}^{L \times d_2}$ is computed as $\mathbf{O} = \text{Softmax}(\mathbf{Q}\mathbf{K}^\top + \mathbf{M})\mathbf{V}$, query, key, and value matrices $\mathbf{Q}, \mathbf{K}, \mathbf{V} = \mathbf{X}\mathbf{W}_{\text{q,k,v}}$ are projected via learnable weights $\boldsymbol{W}_{\text{q,k,v}} \in \mathbb{R}^{d_1 \times d_2}$. The causal mask $\mathbf{M} \in \{-\infty, 0\}^{L \times L}$ ensures that $\mathbf{M}_{ij} = 0$ for $i \geq j$ and $-\infty$ otherwise. While this formulation enables efficient parallel training, inference is computationally demanding when viewed in its recurrent form: $\boldsymbol{o}_t = \sum_{i=1}^t \left(\exp(\boldsymbol{q}_t^\top \boldsymbol{k}_i) / \sum_{j=1}^t \exp(\boldsymbol{q}_t^\top \boldsymbol{k}_j)\right) \boldsymbol{v}_i$, where $\boldsymbol{q}_t, \boldsymbol{k}_t, \boldsymbol{v}_t = \boldsymbol{W}_{\text{q,k,v}}^\top \boldsymbol{x}_t$ represent the vectors for the current token $\boldsymbol{x}_t \in \mathbb{R}^{d_1}$. This mechanism requires $O(L)$ memory per step to store the expanding "KV cache" $\{\boldsymbol{k}_i, \boldsymbol{v}_i\}_{i=1}^t$, leading to an aggregate $O(L^2)$ computational complexity.

**The Linear Attention** circumvents this quadratic cost by linearizing the Softmax operator (Katharopoulos et al., 2020; Kasai et al., 2021; Peng et al., 2021). Removing the Softmax operator yields the output: $\boldsymbol{o}_t = \sum_{i=1}^t (\boldsymbol{q}_t^\top \boldsymbol{k}_i)\boldsymbol{v}_i = (\boldsymbol{q}_t^\top \sum_{i=1}^t \boldsymbol{k}_i \boldsymbol{v}_i^\top)^\top = \mathbf{S}_t^\top \boldsymbol{q}_t$. This reformulates the matrix $\mathbf{S}_t := \sum_{i=1}^t \boldsymbol{k}_i \boldsymbol{v}_i^\top = \mathbf{S}_{t-1} + \boldsymbol{k}_t \boldsymbol{v}_t^\top \in \mathbb{R}^{d_k \times d_v}$ as "Fast Weights" (Hinton & Plaut, 1987; Schmidhuber, 1992; Ba et al., 2016; Schlag et al., 2021; Irie et al., 2021). The fully parallel form of causal linear attention remains quadratic in $L$, is given by $\mathbf{O} = ((\mathbf{Q}\mathbf{K}^\top) \odot \mathbf{M})\mathbf{V}$, where causal mask $\mathbf{M} \in \{0, 1\}^{L \times L}$ is $\mathbf{M}_{ij} = 1$ only when $i \geq j$.

**The chunkwise parallel form** of linear attention optimally balances between fully parallel and recurrent formulations, enabling subquadratic training complexity (Sun et al., 2023). For chunks $t \in [0, \frac{L}{C}]$, the output of each chunk is decomposed as $\mathbf{O}_{[t]} = \mathbf{O}_{[t]}^{\text{inter}} + \mathbf{O}_{[t]}^{\text{intra}}$. The intra-chunk output is computed in parallel as $\mathbf{O}_{[t]}^{\text{intra}} = ((\mathbf{Q}_{[t]}\mathbf{K}_{[t]}^\top) \odot \mathbf{M}) \cdot \mathbf{V}_{[t]}$, while the inter-chunk output is computed as

*Table 1.* Recurrent Associative Memory and Optimization Perspectives. The update rule of linear attention models is the closed-form solution of the corresponding objective function under its specified optimizer. This table mainly follows Team et al. (2025).

| Model | Update Rule (Closed form of $\mathcal{L}$ solved by $\mathcal{O}$) | Loss Objective $\mathcal{L}$ | Optimizer $\mathcal{O}$ |
|---|---|---|---|
| Self Attention | $\mathbf{S}_t.\mathrm{append}(\boldsymbol{k}_t, \boldsymbol{v}_t)$ | - | - |
| Vanilla Linear Attention | $\mathbf{S}_t = \mathbf{S}_{t-1} + \boldsymbol{k}_t \boldsymbol{v}_t^\top$ | $-\langle \mathbf{S}_{t-1}^\top \boldsymbol{k}_t, \boldsymbol{v}_t \rangle$ | SGD |
| Mamba2 (Dao & Gu, 2024) | $\mathbf{S}_t = \alpha_t \mathbf{S}_{t-1} + \beta_t \boldsymbol{k}_t \boldsymbol{v}_t^\top$ | $-\beta_t \langle \mathbf{S}_{t-1}^\top \boldsymbol{k}_t, \boldsymbol{v}_t \rangle + \frac{1}{2} \left\| \sqrt{1-\alpha_t} \mathbf{S}_{t-1} \right\|_F^2$ | SGD |
| GLA (Yang et al., 2024a) | $\mathbf{S}_t = \mathrm{Diag}(\boldsymbol{\alpha}_t) \mathbf{S}_{t-1} + \beta_t \boldsymbol{k}_t \boldsymbol{v}_t^\top$ | $-\beta_t \langle \mathbf{S}_{t-1}^\top \boldsymbol{k}_t, \boldsymbol{v}_t \rangle + \frac{1}{2} \left\| \sqrt{1-\mathrm{Diag}(\alpha_t)} \mathbf{S}_{t-1} \right\|_F^2$ | SGD |
| DeltaNet (Yang et al., 2024b) | $\mathbf{S}_t = \left(\mathbf{I} - \beta_t \boldsymbol{k}_t \boldsymbol{k}_t^\top\right) \mathbf{S}_{t-1} + \beta_t \boldsymbol{k}_t \boldsymbol{v}_t^\top$ | $\frac{\beta_t}{2} \left\| \mathbf{S}_{t-1}^\top \boldsymbol{k}_t - \boldsymbol{v}_t \right\|^2$ | SGD |
| GDN (Yang et al., 2025) | $\mathbf{S}_t = \alpha_t \left(\mathbf{I} - \beta_t \boldsymbol{k}_t \boldsymbol{k}_t^\top\right) \mathbf{S}_{t-1} + \beta_t \boldsymbol{k}_t \boldsymbol{v}_t^\top$ | $\frac{\beta_t}{2} \left\| \tilde{\mathbf{S}}_{t-1}^\top \boldsymbol{k}_t - \boldsymbol{v}_t \right\|^2$ (where $\tilde{\mathbf{S}}_{t-1} = \alpha_t \mathbf{S}_{t-1}$) | SGD |
| RWKV-7 (Peng et al., 2025) | $\mathbf{S}_t = \left(\mathrm{Diag}(\boldsymbol{\alpha}_t) - \gamma_t \boldsymbol{k}_t \boldsymbol{k}_t^\top\right) \mathbf{S}_{t-1} + \beta_t \boldsymbol{k}_t \boldsymbol{v}_t^\top$ | $\frac{\gamma_t}{2} \left\| \mathbf{S}_{t-1}^\top \tilde{\boldsymbol{k}}_t - \boldsymbol{v}_t \right\|^2 + \frac{1}{2} \left\| \sqrt{\mathrm{Diag}\left(1-\alpha_t\right)} \mathbf{S}_{t-1} \right\|_F^2$ | SGD |
| Comba (Hu et al., 2025) | $\mathbf{S}_t = \left(\alpha_t \mathbf{I} - \alpha_t \tilde{\beta}_t \boldsymbol{k}_t \boldsymbol{k}_t^\top\right) \mathbf{S}_{t-1} + \beta_t \boldsymbol{k}_t \boldsymbol{v}_t^\top$ | $\frac{\beta_t}{2} \left\| \mathbf{S}_{t-1}^\top \boldsymbol{k}_t - \boldsymbol{v}_t \right\|^2 + \frac{1}{2} \left\| \sqrt{1-\alpha_t} \mathbf{S}_{t-1} \right\|_F^2$ | SGD |
| KDA (Team et al., 2025) | $\mathbf{S}_t = \mathrm{Diag}(\boldsymbol{\alpha}_t) \left(\mathbf{I} - \beta_t \boldsymbol{k}_t \boldsymbol{k}_t^\top\right) \mathbf{S}_{t-1} + \beta_t \boldsymbol{k}_t \boldsymbol{v}_t^\top$ | $\frac{\beta_t}{2} \left\| \tilde{\mathbf{S}}_{t-1}^\top \boldsymbol{k}_t - \boldsymbol{v}_t \right\|^2$ (where $\tilde{\mathbf{S}}_{t-1} = \mathrm{Diag}(\boldsymbol{\alpha}_t) \mathbf{S}_{t-1}$) | SGD |
| **MDN (Ours)**[1] | $\mathbf{S}_t = \left(\tilde{\alpha}_t \mathbf{I} - \tilde{\beta}_t \boldsymbol{k}_t \boldsymbol{k}_t^\top\right) \mathbf{S}_{t-1} - \tilde{\gamma}_t \mathbf{S}_{t-2} + \tilde{\beta}_t \boldsymbol{k}_t \boldsymbol{v}_t^\top$ | $\frac{\beta_t}{2} \left\| \tilde{\mathbf{S}}_{t-1}^\top \boldsymbol{k}_t - \boldsymbol{v}_t \right\|^2$ (where $\tilde{\mathbf{S}}_{t-1} = \alpha_t \mathbf{S}_{t-1}$) | Momentum GD |

[1] After derivation, where $\tilde{\alpha}_t = \alpha_t + \frac{\mu_t \beta_t}{\beta_{t-1}}$, $\tilde{\beta}_t = \beta_t \eta_t$, $\tilde{\gamma}_t = \frac{\alpha_{t-1} \mu_t \beta_t}{\beta_{t-1}}$. In practice, we still use the stepwise momentum form for recurrent decoding as shown in Eq. (4)-(5).

$\mathbf{O}_{[t]}^{\mathrm{inter}} = \mathbf{Q}_{[t]} \mathbf{S}_{[t]}$. The inter-chunk state is updated recurrently by $\mathbf{S}_{[t+1]} = \mathbf{S}_{[t]} + \sum_{i=tC+1}^{(t+1)C} \boldsymbol{k}_i \boldsymbol{v}_i^\top = \mathbf{S}_{[t]} + \mathbf{K}_{[t]}^\top \mathbf{V}_{[t]}$. This formulation yields an overall training complexity of $O(LCd + Ld^2)$, which is significantly lower than the $O(L^2 d)$ cost of the fully parallel form when $L \gg C$ (Yang et al., 2024a). The chunkwise form recovers the fully parallel case when $C = L$ and the recurrent case when $C = 1$.

## 2.2. Linear Attention with Decay Rule

The vanilla linear attention underperformed Transformers due to the unbounded nature of its cumulative hidden state. To address this, a common solution is to introduce a Decay rule to selectively forget historical information. For example, the recurrence of Mamba2 (Dao & Gu, 2024) as:

$$\mathbf{S}_t = \alpha_t \mathbf{S}_{t-1} + \boldsymbol{k}_t \boldsymbol{v}_t^\top \in \mathbb{R}^{d_k \times d_v}, \quad \boldsymbol{o}_t = \mathbf{S}_t^\top \boldsymbol{q}_t \in \mathbb{R}^{d_v},$$

where scalar decay $\alpha_t \in (0, 1)$ is a data-dependent term that varies with different input. By defining the cumulative product $\bar{\alpha}_j = \prod_{i=1}^{j} \alpha_i$, the decay term can be expressed as both a vector form (left) and a matrix parallel form (right):

$$\boldsymbol{o}_t = \sum_{i=1}^{t} \frac{\bar{\alpha}_t}{\bar{\alpha}_i} \left(\boldsymbol{q}_t^\top \boldsymbol{k}_i\right) \boldsymbol{v}_i, \quad \mathbf{O} = \left(\left(\mathbf{Q}\mathbf{K}^\top\right) \odot \Gamma\right) \mathbf{V},$$

where $\Gamma \in \mathbb{R}^{L \times L}$ is a decay-aware causal mask with $\Gamma_{ij} = \frac{\bar{\alpha}_i}{\bar{\alpha}_j}$ if $i \geq j$ and 0 otherwise. Linear attention with data-dependent decay can be seamlessly extended to a chunkwise algorithm, following the State Space Duality (SSD) framework proposed by Dao & Gu (2024):

$$\mathbf{S}_{[t+1]} = \bar{\alpha}_{[t]}^C \mathbf{S}_{[t]} + \left(\mathrm{Diag}\left(\frac{\bar{\alpha}_{[t]}^C}{\bar{\boldsymbol{\alpha}}_{[t]}}\right) \cdot \mathbf{K}_{[t]}\right)^\top \mathbf{V}_{[t]},$$

$$\mathbf{O}_{[t]} = \mathrm{Diag}(\bar{\boldsymbol{\alpha}}_{[t]}) \cdot \mathbf{Q}_{[t]} \mathbf{S}_{[t]}^\top + \left(\mathbf{Q}_{[t]} \mathbf{K}_{[t]}^\top \odot \Gamma_{[t]}\right) \cdot \mathbf{V}_{[t]},$$

where mask $(\Gamma_{[t]})_{ij} = \frac{\bar{\alpha}_{[t]}^i}{\bar{\alpha}_{[t]}^j}$ for $i \geq j$, $\bar{\alpha}_{[t]}^j = \prod_{i=tC+1}^{tC+j} \alpha_i$.

When $\alpha_t$ reformulate as data-independent scalar $\alpha$, the formulation becomes RetNet (Sun et al., 2023) and Lightning Attention (Qin et al., 2024a). Furthermore, scalar-valued $\alpha_t$ can be extended to be vector-valued $\boldsymbol{\alpha}_t$ for more fine-grained decay, where efficient chunkwise training algorithms were proposed by GLA (Yang et al., 2024a) and subsequently adopted in Qin et al. (2024b); Zhang et al. (2024); Chou et al. (2024); He et al. (2024); Lu et al. (2025).

## 2.3. Linear Attention with Delta Rule

The Gated DeltaNet (GDN) (Yang et al., 2025) further improves the mamba2 by incorporating the Delta rule (Schlag et al., 2021), which dynamically updates the value ($\boldsymbol{v}_t$) associated with the input key ($\boldsymbol{k}_t$) to generate a new correction value ($\boldsymbol{v}_t^{\mathrm{new}}$) based on the input gate $\beta_t \in (0, 1)$.

$$\mathbf{S}_t = \alpha_t \mathbf{S}_{t-1} + \beta_t \boldsymbol{k}_t \underbrace{\left(\boldsymbol{v}_t - \alpha_t \mathbf{S}_{t-1}^\top \boldsymbol{k}_t\right)^\top}_{\text{Updated } \boldsymbol{v}_t^{\mathrm{new}}}$$

$$= \alpha_t \left(\mathbf{I} - \beta_t \boldsymbol{k}_t \boldsymbol{k}_t^\top\right) \mathbf{S}_{t-1} + \beta_t \boldsymbol{k}_t \boldsymbol{v}_t^\top.$$

Despite demonstrating superior associative recall, the Delta rule had remained computationally challenging until Yang et al. (2024b) introduced an efficient chunkwise algorithm.

Specifically, expanding the recurrence reveals the cumulative products of generalized Householder transition matrices $\prod_t \left(\mathbf{I} - \beta_t \boldsymbol{k}_t \boldsymbol{k}_t^\top\right) \in \mathbb{R}^{d_k \times d_k}$, which are optimized via the WY representation (Bischof & Loan, 1985) to produce the efficient chunkwise computation (Yang et al., 2025),

$$\mathbf{S}_{[t+1]} = \bar{\alpha}_{[t]}^C \mathbf{S}_{[t]} + \left(\mathrm{Diag}\left(\frac{\bar{\alpha}_{[t]}^C}{\bar{\boldsymbol{\alpha}}_{[t]}}\right) \cdot \mathbf{K}_{[t]}\right)^\top \widetilde{\mathbf{V}}_{[t]},$$

$$\mathbf{O}_{[t]} = \mathrm{Diag}(\bar{\boldsymbol{\alpha}}_{[t]}) \cdot \mathbf{Q}_{[t]} \mathbf{S}_{[t]}^\top + \left(\mathbf{Q}_{[t]} \mathbf{K}_{[t]}^\top \odot \Gamma_{[t]}\right) \cdot \widetilde{\mathbf{V}}_{[t]}.$$

The core difference from Mamba-2 lies in the correction term of correction value $\widetilde{\mathbf{V}}_{[t]} = \mathbf{U}_{[t]} - \mathbf{W}_{[t]}\mathbf{S}_{[t]}$. The chunked matrix $\mathbf{W}_{[t]}$ and $\mathbf{U}_{[t]}$ are obtained by the UT transform (Joffrain et al., 2006) as deduced by Yang et al. (2025):

$$\mathbf{U}_{[t]} = \mathbf{T}_{[t]}\mathbf{V}_{[t]} \in \mathbb{R}^{C \times d_v}, \quad \mathbf{W}_{[t]} = \mathbf{T}_{[t]}\mathbf{K}_{[t]} \in \mathbb{R}^{C \times d_k},$$

$$\mathbf{T}_{[t]} = \mathrm{Diag}(\boldsymbol{\beta}_{[t]}) \left( \mathbf{I} + \left( \mathrm{Diag}(\boldsymbol{\beta}_{[t]})\mathbf{K}_{[t]}\mathbf{K}_{[t]}^\top \right) \odot \mathbf{M}_{[t]}^- \right)^{-1},$$

where $\mathbf{T}_{[t]}, \mathbf{M}_{[t]}^- \in \mathbb{R}^{C \times C}$ is lower triangular matrix. Further advancements, such as KDA (Team et al., 2025), extend the delta gating $\alpha_t$ to a vector-valued $\boldsymbol{\alpha}_t$, while Comba (Hu et al., 2025) introduces a closed-loop correction to further enhance GDN. We provide additional related work in § B.

## 3. Method

To incorporate the Stepwise Momentum mechanism into Linear Attention, we first derive its recurrent update and then develop an exact chunkwise parallel formulation. By characterizing the momentum rule as a second-order dynamical system, we obtain a spectral perspective that facilitates stability analysis and guides the design of robust gating constraints. Finally, we present Momentum DeltaNet (MDN), a high-performance architecture that combines the stepwise momentum rule with an effective spectral gating constraint.

### 3.1. Linear Attention with Stepwise Momentum Rule

In this section, we construct both the recurrent update and the chunkwise parallel form for the Stepwise Momentum mechanism. Consider an optimizer with momentum state $\mathbf{M}_t$, decay factor $\mu_t$ and learning rate $\beta_t$ (where $\eta_t$ is a scaling factor):

$$\mathbf{M}_t = \mu_t \cdot \mathbf{M}_{t-1} + \eta_t \cdot \nabla \mathcal{L}(\widetilde{\mathbf{S}}_{t-1}), \quad (1)$$

$$\mathbf{S}_t = \widetilde{\mathbf{S}}_{t-1} - \beta_t \cdot \mathbf{M}_t. \quad (2)$$

The learning objective is expected the key $\boldsymbol{k}_t$ can associate the memory of the corresponding value $\boldsymbol{v}_t$ from the decayed fast weight $\widetilde{\mathbf{S}}_{t-1} = \alpha_t \mathbf{S}_{t-1}$. Defining the loss as:

$$\mathcal{L}(\widetilde{\mathbf{S}}_{t-1}) = \frac{1}{2}\|\boldsymbol{v}_t - \widetilde{\mathbf{S}}_{t-1}^\top \boldsymbol{k}_t\|_2^2, \quad (3)$$

the gradient with respect to the fast weight is $\nabla_{\widetilde{\mathbf{S}}}\mathcal{L}(\widetilde{\mathbf{S}}_{t-1}) = -\boldsymbol{k}_t(\boldsymbol{v}_t - \widetilde{\mathbf{S}}_{t-1}^\top \boldsymbol{k}_t)^\top$, which yields the recurrence:

$$\mathbf{M}_t = \mu_t \cdot \mathbf{M}_{t-1} - \eta_t \cdot \boldsymbol{k}_t(\boldsymbol{v}_t - \alpha_t \cdot \mathbf{S}_{t-1}^\top \boldsymbol{k}_t)^\top, \quad (4)$$

$$\mathbf{S}_t = \alpha_t \cdot \mathbf{S}_{t-1} - \beta_t \cdot \mathbf{M}_t, \quad (5)$$

where the fast weight and momentum are $\mathbf{S}_t, \mathbf{M}_t \in \mathbb{R}^{d_k \times d_v}$. The output is queried from the fast weight as $\boldsymbol{o}_t = \mathbf{S}_t^\top \boldsymbol{q}_t$.

Under the test-time training (TTT) interpretation, Eq. (4)–(5) provides a unified recurrence family for linear attention.

Here, $\boldsymbol{k}_t$ acts as the input to a fast weight memory, and the update is driven by a prediction error (correction term). We define the correction as $\widetilde{\boldsymbol{v}}_t := \boldsymbol{v}_t - \mathbf{S}_{t-1}^\top \boldsymbol{p}_t$, where we set $\boldsymbol{p}_t = \alpha_t \boldsymbol{k}_t$ inspired by Hu et al. (2025). As special cases, setting $\mu_t = 0$ and $\eta_t = 1$ recovers first order updates: with $\boldsymbol{p}_t = \alpha_t \boldsymbol{k}_t$ the recurrence matches Gated DeltaNet; with $\alpha_t = 1$ and $\boldsymbol{p}_t = \boldsymbol{k}_t$ it reduces to DeltaNet; and with $\boldsymbol{p}_t = \mathbf{0}$ it reduces to a decay-style update. These recurrences can be interpreted as closed-form online optimization steps under different latent objectives (Table 1).

**Parallel Formulation.** Then we consider the Momentum with $\mu \neq 0$ to derivative the parallel formulation. To expand the recurrent form as follows by assuming already know the correction value $\widetilde{\boldsymbol{v}}_t$, expanding the $\mathbf{M}_t = \mu_t \mathbf{M}_{t-1} - \boldsymbol{k}_t \widetilde{\boldsymbol{v}}_t^\top$ [1], we can obtain the $\mathbf{M}_t$ general parallel form in Eq. (6):

$$\mathbf{M}_t = \bar{\mu}_t \mathbf{M}_0 - \left( \mathrm{Diag}\left( \frac{\bar{\mu}_t}{\bar{\boldsymbol{\mu}}} \right) \cdot \mathbf{K} \right)^\top \widetilde{\mathbf{V}}, \quad (6)$$

Substituting the expanded momentum $\mathbf{M}_t$ from Eq. (6) into the $\mathbf{S}_t$ in Eq. (5) yields the expanded form of $\mathbf{S}_t$:

$$\mathbf{S}_t = \bar{\alpha}_t \mathbf{S}_0 - \sum_{i=1}^{t} \beta_i \frac{\bar{\alpha}_t}{\bar{\alpha}_i} \bar{\mu}_i \mathbf{M}_0 + \sum_{i=1}^{t} \beta_i \frac{\bar{\alpha}_t}{\bar{\alpha}_i} \sum_{j=1}^{i} \frac{\bar{\mu}_i}{\bar{\mu}_j} \boldsymbol{k}_j \widetilde{\boldsymbol{v}}_j^\top.$$

However, the nested summation initially obstructs direct parallelization. Our strategy is to decouple the coefficient and outer product by the transformation as shown in Eq. (7):

$$\sum_{i=1}^{t} \sum_{j=1}^{i} a_i \cdot b_j = \sum_{j=1}^{t} \sum_{i=j}^{t} a_i \cdot b_j = \sum_{i=1}^{t} \sum_{j=i}^{t} a_j \cdot b_i, \quad (7)$$

where the equality follows by viewing the nested summation as a traversal over the same lower-triangular index domain and reordering it from a row-wise to a column-wise scan.

Applying the key transformation Eq. (7) on $\mathbf{S}_t$, then we can decouple the coefficient from the nested summation to get:

$$\sum_{i=1}^{t} \beta_i \frac{\bar{\alpha}_t}{\bar{\alpha}_i} \sum_{j=1}^{i} \frac{\bar{\mu}_i}{\bar{\mu}_j} \boldsymbol{k}_j \widetilde{\boldsymbol{v}}_j^\top = \sum_{i=1}^{t} \frac{\bar{\alpha}_t}{\bar{\mu}_i} \left( \sum_{j=i}^{t} \frac{\beta_j \bar{\mu}_j}{\bar{\alpha}_j} \right) \boldsymbol{k}_i \widetilde{\boldsymbol{v}}_i^\top,$$

then we get the new parallel formulation as Eq. (8),

$$\mathbf{S}_t = \bar{\alpha}_t \mathbf{S}_0 - b_t \mathbf{M}_0 + \left( \mathrm{Diag}\left( \frac{\gamma_{t,t}}{\gamma_{t,:}} \right) \cdot \mathbf{K} \right)^\top \widetilde{\mathbf{V}}. \quad (8)$$

As shown in Eq. (8), the fast weight is the function of the initial state and momentum and the decoupled coefficients. where the corresponding coefficients are defined as below,

$$\bar{\mu}_t := \prod_{j=1}^{t} \mu_j, \quad \bar{\alpha}_t := \prod_{j=1}^{t} \alpha_j, \quad c_t := \sum_{i=1}^{t} \frac{\beta_i \bar{\mu}_i}{\bar{\alpha}_i}, \quad (9)$$

$$b_t := \bar{\alpha}_t c_t, \quad \gamma_{t,i} := \frac{\bar{\alpha}_t}{\bar{\mu}_i}(c_t - c_{i-1}) \quad \text{for} \quad i <= t.$$

---

[1] We can ignore the $\eta_t$ due to this scalar can be absorbed by $\boldsymbol{k}_t$, same the scalar $\alpha_t$ also can absorbed in $\boldsymbol{p}_t$.

Then, the challenge of how to realize the efficient parallel formulation now shifts to how to efficiently compute these coefficients (More details of parallel derivation see § C).

**Coefficient Chunkwise.** The coefficients in Eq. (9) can be computed in chunkwise parallel within the log-domain,

$$\bar{\boldsymbol{\mu}}_{[t]}^{\log} = \text{cumsum}(\boldsymbol{\mu}_{[t]}^{\log}), \quad \bar{\boldsymbol{\alpha}}_{[t]}^{\log} = \text{cumsum}(\boldsymbol{\alpha}_{[t]}^{\log}) \quad \in \mathbb{R}^C,$$

$$\boldsymbol{c}_{[t]}^{\log} = \log(\text{cumsum}(\exp(\boldsymbol{\beta}_{[t]}^{\log} + \bar{\boldsymbol{\mu}}_{[t]}^{\log} - \bar{\boldsymbol{\alpha}}_{[t]}^{\log}))) \quad \in \mathbb{R}^C.$$

Here, the cumsum denotes the operator of the Prefix Sum algorithm applied within each chunk with $O(\log C)$ complexity. Furthermore, the log-cumsum-exp operator can be safely computed with $O(1)$ time complexity and acceptable $O(C^2)$ space complexity for each chunk in parallel, due to the chunk size $C$ is the small fixed constant (More detail in the § D with Algorithm 1). Further, the chunk form of $b_t$ and the $\gamma_{t,i}$ are computed as following Eq.(10) and (11),

$$\boldsymbol{b}_{[t]} = \exp(\boldsymbol{\mu}_{[t]}^{\log} + \boldsymbol{c}_{[t]}^{\log}) \qquad \in \mathbb{R}^C, \quad (10)$$

$$\boldsymbol{\Gamma}_{[t]} = \exp(\widetilde{\mathbf{A}}_{[t]}^{\log}) \odot \left(1 - \exp(\mathbf{S}_{[t]}^{\log})\right) \quad \in \mathbb{R}^{C \times C}, \quad (11)$$

where the chunk matrix $\widetilde{\mathbf{A}}_{[t]}^{\log}, \mathbf{S}_{[t]}^{\log} \in \mathbb{R}^{C \times C}$ is computed,

$$(\widetilde{\mathbf{A}}_{[t]}^{\log})_{ij} = (\bar{\boldsymbol{\alpha}}_{[t]}^{\log} + \boldsymbol{c}_{[t]}^{\log})_i - (\bar{\boldsymbol{\mu}}_{[t]}^{\log})_j \qquad \text{for } i \geq j, \quad (12)$$

$$(\mathbf{S}_{[t]}^{\log})_{ij} = (\boldsymbol{c}_{[t]}^{\log})_{j-1} - (\boldsymbol{c}_{[t]}^{\log})_i \qquad \text{for } i \geq j. \quad (13)$$

These lower triangular matrices are computed by broadcasting the chunkwise vectors as shown in Eq. (12) and (13). The explicit separation of $\mathbf{S}_{[t]}^{\log}$ is intended to maintain numerical stability and avoid $\log(0)$ in the log-domain.

**Chunkwise Algorithm.** Subsequently, we can extend the parallel formulation to the chunkwise algorithm as,

$$\mathbf{O}_{[t]} = \mathbf{O}_{[t]}^{\text{Inter}} + \mathbf{O}_{[t]}^{\text{Intra}} \qquad \in \mathbb{R}^{C \times d_v}, \quad (14)$$

where the inter-chunk output $\mathbf{O}_{[t]}^{\text{Inter}}$ and intra-chunk output $\mathbf{O}_{[t]}^{\text{Intra}}$ are computed as follows:

$$\mathbf{O}_{[t]}^{\text{Inter}} = \text{Diag}(\bar{\boldsymbol{\alpha}}_{[t]}) \cdot \mathbf{Q}_{[t]} \mathbf{S}_{[t]} - \text{Diag}(\boldsymbol{b}_{[t]}) \cdot \mathbf{Q}_{[t]} \mathbf{M}_{[t]},$$

$$\mathbf{O}_{[t]}^{\text{Intra}} = \left((\mathbf{Q}_{[t]} \mathbf{K}_{[t]}^\top) \odot \boldsymbol{\Gamma}_{[t]}\right) \cdot \widetilde{\mathbf{V}}_{[t]}.$$

The hidden states of each chunk are updated following:

$$\mathbf{M}_{[t+1]} = \bar{\boldsymbol{\mu}}_{[t]}^C \cdot \mathbf{M}_{[t]} - \left(\text{Diag}\left(\frac{\bar{\boldsymbol{\mu}}_{[t]}^C}{\bar{\boldsymbol{\mu}}_{[t]}}\right) \cdot \mathbf{K}_{[t]}\right)^\top \widetilde{\mathbf{V}}_{[t]},$$

$$\mathbf{S}_{[t+1]} = \bar{\boldsymbol{\alpha}}_{[t]}^C \cdot \mathbf{S}_{[t]} - b_{[t]}^C \cdot \mathbf{M}_{[t]} + \left(\text{Diag}\left(\boldsymbol{\Gamma}_{[t]}^C\right) \cdot \mathbf{K}_{[t]}\right)^\top \widetilde{\mathbf{V}}_{[t]},$$

where $\boldsymbol{\Gamma}_{[t]}^C$ is the $C$-th row (last row) vector of the $t$-th chunk of causal mask $\boldsymbol{\Gamma}_{[t]} \in \mathbb{R}^{C \times C}$, and $\widetilde{\mathbf{V}}_{[t]}$ is computed as,

$$\widetilde{\mathbf{V}}_{[t]} = \mathbf{U}_{[t]} - \mathbf{Y}_{[t]} \mathbf{S}_{[t]} + \mathbf{Z}_{[t]} \mathbf{M}_{[t]} \quad \in \mathbb{R}^{C \times d_v},$$

where $\mathbf{U}_{[t]} \in \mathbb{R}^{C \times d_v}$ as shown in Eq. (15) and $\mathbf{Y}_{[t]}, \mathbf{Z}_{[t]} \in \mathbb{R}^{C \times d_k}$ (Eq. (16) and (17)) are computed by $\mathbf{T}_{[t]} \in \mathbb{R}^{C \times C}$:

$$\mathbf{U}_{[t]} = \mathbf{T}_{[t]} \cdot \mathbf{V}_{[t]}, \quad (15)$$

$$\mathbf{Y}_{[t]} = \mathbf{T}_{[t]} \cdot \left(\text{Diag}(\bar{\boldsymbol{\alpha}}_{[t]}^{0 \to C-1}) \cdot \mathbf{P}_{[t]}\right), \quad (16)$$

$$\mathbf{Z}_{[t]} = \mathbf{T}_{[t]} \cdot (\text{Diag}(\boldsymbol{b}_{[t]}^{0 \to C-1}) \cdot \mathbf{P}_{[t]}), \quad (17)$$

$$\mathbf{T}_{[t]} = \text{Tril}\left(\mathbf{I}_{[t]} + \left(\mathbf{P}_{[t]} \mathbf{K}_{[t]}^\top \odot \boldsymbol{\Gamma}_{[t]}^-\right)\right)^{-1}, \quad (18)$$

where $\boldsymbol{\Gamma}_{[t]}^-$ denotes the strictly lower triangular part of the mask obtained from Eq. (11). The detailed recurrent and chunkwise parallel PyTorch-style codes are provided in § E.

**Practical Considerations.** In the Triton implementation, Comba and GDN recompute $\mathbf{S}_{[t]}$ for each chunk during the backward pass to conserve memory. However, directly extending this approach to chunkwise algorithm for momentum is inefficient, as it necessitates the recomputation of both the hidden state $\mathbf{S}_{[t]}$ and the momentum state $\mathbf{M}_{[t]}$. To address this, we materialize the correction value $\widetilde{\mathbf{V}}_{[t]}$. During the forward pass, we compute the inter-chunk output and $\widetilde{\mathbf{V}}_{[t]}$ without storing the full states $\mathbf{S}_{[t]}$ or $\mathbf{M}_{[t]}$. In the backward pass, these states are efficiently reconstructed from $\widetilde{\mathbf{V}}_{[t]}$ for gradient computation. This strategy improves training throughput with minimal memory overhead (§ 4.3).

### 3.2. Eigenvalue Analysis and Discussion

To analyze the representation capacity of the proposed mechanism, we reformulate its recurrence as a linear dynamical system and study the eigenvalues of the transition matrix $\mathbf{A}_t$ (Eq. 19). While previous models rely on discrete first-order dynamics, the momentum rule evolves into a second-order system, expanding the eigenvalue space (Figure 2):

$$\mathbf{S}_t = \mathbf{A}_t \mathbf{S}_{t-1}, \quad \begin{pmatrix} \mathbf{S}_t \\ \mathbf{M}_t \end{pmatrix} = \mathbf{A}_t \begin{pmatrix} \mathbf{S}_{t-1} \\ \mathbf{M}_{t-1} \end{pmatrix}. \quad (19)$$

**Limitations of First Order Dynamics.** For conventional 1st-order systems (e.g., decay and delta rules) as shown in Eq. (19)(left), the eigenvalues of $\mathbf{A}_t$ lie on the real axis under standard parameterizations. While different mechanism construct distinct $\mathbf{A}_t$, both are constrained to maintain $|\rho(\mathbf{A}_t)| \leq 1$. For the decay rule with $\mathbf{A}_t = \alpha_t \mathbf{I}$, the standard gating $\alpha_t \in (0, 1)$ ensures $\rho(\mathbf{A}_t) = \alpha \in (0, 1)$. The delta rule constructs an IPLR structure $\mathbf{A}_t = \alpha_t (\mathbf{I} - \beta_t \boldsymbol{k}_t \boldsymbol{k}_t^\top)$. Under the key normalization[2] ($\|\boldsymbol{k}_t\|^2 = 1$), the spectral radius $\rho(\mathbf{A}_t) = \alpha_t (1 - \beta_t) \in (0, 1)$ are restricted to the positive real axis with the $\beta_t \in (0, 1)$ as shown in Figure 2(a). Further, Grazzi et al. (2024) and Siems et al. (2025) relax $\beta \in (0, 2)$, achieving negative eigenvalues (sign-flipping) to enable state tracking. More general DPLR

---

[2] Lei et al. (2025) relax the key normalization by $\widetilde{\beta} = \beta / \|\boldsymbol{k}_t\|^2$.

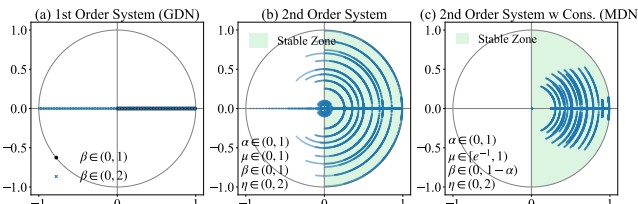

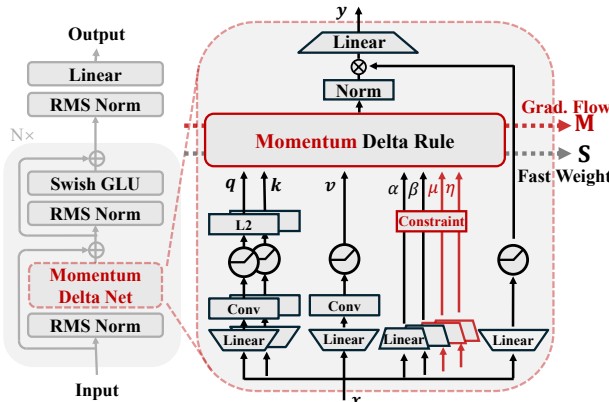

*Figure 2.* Spectral root trajectories of $\mathbf{A}_t$ by sweeping coefficients. (a) Roots lie on the real axis $\lambda = \alpha(1-\beta)$, where $\beta \in (0,1)$ yields positive value eigenvalues, while $\beta \in (1,2)$ produces sign-flipping modes in negative value eigenvalues. (b) The $\alpha, \mu, \beta \in (0,1)$ and $\eta \in (0,2)$ yields a two-dimensional spectral region that may enter the left half-plane. (c) With the example constraint $\beta < 1-\alpha$ and $\mu \in [e^{-1}, 1)$, all roots strictly confined to the right half-plane.

and SPLR[3] structures $\mathbf{A}_t = \alpha_t \mathbf{I} - \beta_t \mathbf{k}_t \mathbf{k}_t^\top$ similarly constrain the eigenvalues to interval $(-1, 1)$. Despite these improvements, these systems remain limited to the real domain. This restriction prevents the system from capturing oscillatory dependencies.

**Second Order Dynamics and Expressivity.** The stepwise momentum rule breaks this real-valued limitation by inducing a second-order system that admits complex conjugate eigenvalues. Sweeping the coefficients produces eigenvalues of the transition matrix $\mathbf{A}_t$ as shown in Figure 2(b) (see § F for a detailed derivation of $\mathbf{A}_t$). First-order systems are restricted to real-valued decay dynamics, whereas second-order systems can admit complex eigenvalues, thereby allowing damped oscillatory behavior. These oscillatory modes expand the expressive capacity of the state space by enabling phase-aware memory. From an optimization perspective, momentum accumulates historical gradients, suppressing high-frequency noise while reinforcing consistent directional signals over the sequence.

**Stability via Quadrant Constraint.** Despite the enhanced expressivity, unconstrained 2nd-order coefficients can trigger catastrophic numerical failures (e.g., NaNs) during training. We attribute this primarily to sign flipping behavior (Goh, 2017) induced by eigenvalues with negative real parts, corresponding to the 2nd and 3rd quadrants. Such modes introduce phase mismatched feedback that disrupts the synergy between fast weights and momentum, leading to destructive interference and transient amplification, even when the spectral radius satisfies $\rho(\mathbf{A}_t) < 1$. To ensure robust large scale training, we constrain the gating mechanism to ensure that eigenvalues lie in the 1st and 4th quadrants (Figure 2c). By enforcing $\beta_t \leq 1 - \alpha_t$ and $\mu_t \in (e^{-1}, 1)$, the system avoids divergent sign-flipping while preserving damped oscillations or decay essential for stable dynamics.

---

[3]The IPLR, DPLR and SPLR denote Identity, Diagonal and Scalar Plus Low Rank, respectively. (Yang et al., 2025; Hu et al., 2025; Peng et al., 2025)

*Figure 3.* The schematic illustration of the MDN architecture, the difference are highlighted in red color.

### 3.3. Neural Architecture

Building upon the stepwise momentum rule, we introduce a stability-aware gating parameterization that yields a linear architecture balancing expressivity and numerical robustness. The overall architecture of Momentum DeltaNet (MDN) is detailed in Figure 3.

The main backbone of our model architecture follows GDN (Yang et al., 2025) and Comba (Hu et al., 2025). Before the output projection through $\mathbf{W}_o \in \mathbb{R}^{d \times d}$, we employ head-wise RMSNorm (Zhang & Sennrich, 2019) and a data-dependent gating mechanism (Qiu et al., 2025) as:

$$\boldsymbol{o}_t = \mathrm{MDN}\left(\boldsymbol{q}_t, \boldsymbol{k}_t, \boldsymbol{v}_t, \alpha_t, \beta_t, \mu_t, \eta_t, \right),$$
$$\boldsymbol{y}_t = \boldsymbol{W}_o\left(\mathrm{Sigmoid}\left(\boldsymbol{W}_g \boldsymbol{x}_t\right) \odot \mathrm{RMSNorm}(\boldsymbol{o}_t)\right),$$

where MDN implements the momentum delta rule, using the chunkwise parallel algorithm for training and the recurrent formulation (Eq. (4)–(5)) for autoregressive decoding. The $\boldsymbol{x}_t \in \mathbb{R}^d$ is the $t$-th token input representation, the input to MDN for each head $h$ is computed as follows,

$$\boldsymbol{q}_t^h, \boldsymbol{k}_t^h = \mathrm{L2Norm}(\mathrm{Silu}(\mathrm{ShortConv}(\boldsymbol{W}_{q/k}^h \boldsymbol{x}_t))) \in \mathbb{R}^{d_k},$$
$$\boldsymbol{v}_t^h = \mathrm{Silu}(\mathrm{ShortConv}(\boldsymbol{W}_v^h \boldsymbol{x}_t)) \in \mathbb{R}^{d_v},$$

where $d_k$ and $d_v$ represent the key and value head dimensions, respectively. For $\boldsymbol{q}, \boldsymbol{k}, \boldsymbol{v}$, we apply a ShortConv followed by a $\mathrm{Silu}(x) = x \cdot \mathrm{Sigmoid}(x)$ activation. We use the output correction with $\boldsymbol{q}_t = \boldsymbol{q}_t - d\boldsymbol{k}_t$ before L2Norm as proposed by Hu et al. (2025). The L2Norm ensures eigenvalue stability, as suggested by Yang et al. (2024b).

**Stability Aware Gating Parameterization.** To promote stable dynamics and bias the eigenvalues of the second-order transition matrix $\mathbf{A}_t$ toward the stable right-half plane (analyzed in § 3.2), we parameterize the gating as:

$$\alpha_t^{\mathrm{log}} = f(\boldsymbol{W}_\alpha \boldsymbol{x}_t) + \alpha_{\mathrm{max}}^{\mathrm{log}}, \quad \mu_t^{\mathrm{log}} = f(\boldsymbol{W}_\mu \boldsymbol{x}_t),$$
$$\beta_t = \sigma(\boldsymbol{W}_\beta \boldsymbol{x}_t) \cdot \beta_{\mathrm{max}}, \quad \eta_t = \tanh(\boldsymbol{W}_\eta \boldsymbol{x}_t / \tau) + 1,$$

$$\alpha_{\max} = \cos^2(\theta_t), \;\; \beta_{\max} = \sin^2(\theta_t), \;\; \theta_t = \arctan(\eta_t \cdot s),$$

where the red part is the differences from GDN. The trainable matrix $W_{\alpha/\beta/\mu/\eta} \in \mathbb{R}^{d_{\text{in}} \times h}$ with $h$ (head number) $\ll$ $d_{\text{in}}$ (input dimension), thus introduces only a negligible parameter overhead. The decay function $f(\boldsymbol{x}_t) = -a \cdot$ softplus$(\boldsymbol{x}_t + b)$ is the same with GDN (Yang et al., 2025) and Mamba2 (Dao & Gu, 2024). $\sigma$ denotes the Sigmoid function. We clamp the minimal value (default with -1) of $\mu^{\log}$ to avoid being too small cause the momentum vanishes. The function $\tanh(\cdot) + 1 \in (0, 2)$ makes sure the mean of $\eta_t$ is close to 1, where the temperature $\tau \geq 1$ for controlling the divergence where we default set $\tau = \sqrt{d_{\text{in}}/h}$, where the scalar $s$ is a scaling factor to control the maximum of $\theta$. The upper bound constraint make sure by $\alpha_{\max} + \beta_{\max} = 1$.

## 4. Experiments

We first evaluate Momentum DeltaNet (MDN) on synthetic benchmarks using the MQAR task to assess in-context retrieval ability. We then scale the model to 400M and 1.3B parameters and evaluate its performance on downstream benchmarks covering commonsense reasoning, retrieval, and long-context modeling. Finally, we analyze the efficiency of the chunkwise algorithm and conduct ablation studies to isolate the contributions of the momentum.

**Baseline.** We evaluate MDN against Transformer (Touvron et al., 2021) and four Linear Attention baselines: Mamba2 (Dao & Gu, 2024), Gated DeltaNet (Yang et al., 2025), Comba (Hu et al., 2025) and Kimi Linear Attention (Team et al., 2025). Transformer is the LLaMA architecture with Rotary Positional Embeddings (Su et al., 2024), SwiGLU (Shazeer, 2020), and RMSNorm (Zhang & Sennrich, 2019). All our baselines are trained for the exact same number of tokens on the same dataset for fair comparison. The more experiment details are provided in § G.

### 4.1. Synthetic Benchmark

We first evaluate the in-context retrieval capabilities of linear attention models using the Multi-Query Associative Recall (MQAR) task, which is highly predictive of language modeling performance (Arora et al., 2023a). In this task, the model must memorize a sequence of key-value pairs and subsequently retrieve the correct values for multiple queries. Formally, given an input sequence $\mathbf{X} = [\boldsymbol{k}_1, \boldsymbol{v}_1, \ldots, \boldsymbol{k}_n, \boldsymbol{v}_n, \texttt{<SEP>}, \boldsymbol{q}_1, \ldots, \boldsymbol{q}_m]$, the model is required to autoregressively predict $\mathbf{Y} = [\boldsymbol{y}_1, \ldots, \boldsymbol{y}_m]$, where each $\boldsymbol{y}_j$ is the value previously associated with query $\boldsymbol{q}_j$ seen earlier. An illustrative example follows:

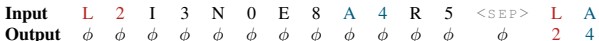

| Input | L | 2 | I | 3 | N | 0 | E | 8 | A | 4 | R | 5 | <SEP> | L | A |
|-------|---|---|---|---|---|---|---|---|---|---|---|---|-------|---|---|
| Output | $\phi$ | $\phi$ | $\phi$ | $\phi$ | $\phi$ | $\phi$ | $\phi$ | $\phi$ | $\phi$ | $\phi$ | $\phi$ | $\phi$ | $\phi$ | 2 | 4 |

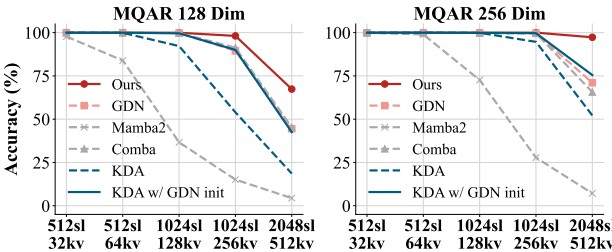

*Figure 4.* MQAR testing result, sl: sequence length, kv: the number of kv pairs. The model with 128 and 256 dimensions.

Models are trained on sequences of up to 256 tokens with 4–64 key value pairs and evaluated under longer contexts ranging from 256 to 2k tokens. As shown in Figure 4, MDN achieves strong retrieval accuracy across a wide range of sequence lengths, performing competitively with KDA, a model specifically optimized for associative recall.

### 4.2. Language Modeling

All models are trained from scratch under identical settings for 400M and 1.3B scale with 4k sequence length. We utilize the AdamW optimizer (Loshchilov & Hutter, 2017) across all experiments. For the 400M/1.3B models, training is conducted on 15B/100B tokens with a 0.5M/1M batch size, respectively. The learning rate follows a cosine schedule, peaking at $3 \times 10^{-4}$ after a warmup phase (0.5B/1B for 400M/1.3B model) and concluding at $3 \times 10^{-5}$. We use the GPT-2 tokenizer on a 100B subset of SlimPajama (Soboleva et al., 2023), which originally contains 627B tokens.

The evaluation mainly follows GND and Comba, containing the Commonsense Reasoning Tasks, In-Context Retrieval Tasks, Long Context Modeling and Needle-In-A-Haystack. The more datasets and setting details are provided in § G.

**Main Results.** As shown in Table 2 (left), recurrent models generally outperform Transformers in perplexity and commonsense reasoning. Notably, MDN achieves the strongest average reasoning performance while maintaining competitive perplexity, indicating that stepwise momentum improves both predictive accuracy and reasoning robustness.

**In-context Retrieval.** As shown in the right half of Table 2, recurrent models typically exhibit degraded recall due to their finite memory states, in contrast to the unbounded context of Transformers. MDN substantially narrows this gap and consistently outperforms other linear baselines.

**Long-context Modeling Ability.** We evaluate long-context performance on LongBench (Bai et al., 2024) using the 1.3B parameter models. As shown in Table 3, MDN achieves the highest average score, with particularly strong improvements on Code and Summarization tasks, demonstrating its effectiveness in long-context reasoning.

*Table 2.* Downstream Tasks Evaluation. The symbol "acc_n" denotes length-normalized accuracy. The commonsense reasoning tasks are performed using the LM evaluation harness (Gao et al., 2024). The in-context retrieval intensive task follows prefix-linear-attention (Arora et al., 2024) with 2K input tokens. All models are implemented and trained using the default configurations provided by the FLA (Yang & Zhang, 2024) and FLAME (Zhang & Yang, 2025) frameworks, respectively. Since KDA architecture is tailored for hybrid models, its pure linear model's result is just for reference. Bold and underlining indicate the **best** and second-best results for linear models, respectively.

| Model | Perplexity | | Commonsense Reasoning Task | | | | | | | | | In-context Retrieval Task | | | | | | |
|---|---|---|---|---|---|---|---|---|---|---|---|---|---|---|---|---|---|---|
| | Lamb. ppl ↓ | Wiki. ppl ↓ | Hella. acc_n ↑ | Lamb. acc ↑ | ARCe acc ↑ | ARCc acc_n ↑ | PIQA acc ↑ | Wino. acc ↑ | BoolQ acc ↑ | SciQ acc ↑ | Avg. acc ↑ | FDA acc ↑ | SWDE acc ↑ | SQD. acc ↑ | NQ acc ↑ | TQA. acc ↑ | Drop acc ↑ | Avg. acc ↑ |
| *400M parameters model with 15B training tokens and 0.5M batch size tokens* | | | | | | | | | | | | | | | | | | |
| Transformer | 54.36 | 32.80 | 34.40 | 33.24 | 45.62 | 24.23 | 64.42 | 52.17 | 59.48 | 70.90 | 48.06 | 43.32 | 31.87 | 29.66 | 17.96 | 41.59 | 18.11 | 30.42 |
| Mamba2 | 60.42 | 33.45 | 35.08 | 29.69 | 46.68 | 23.55 | 65.18 | 52.09 | 59.14 | 71.40 | 47.85 | 11.81 | 17.24 | 27.01 | 13.78 | 38.92 | 17.97 | 21.12 |
| GDN | 45.63 | 32.10 | 34.90 | 34.85 | 46.13 | 24.91 | 65.56 | 52.33 | 57.86 | 71.50 | 48.51 | 14.99 | 20.99 | 27.24 | 14.76 | 40.88 | 18.69 | 22.93 |
| Comba | 46.19 | 31.73 | 35.78 | 34.31 | 47.05 | 24.66 | 65.78 | 51.54 | 58.32 | 73.80 | 48.91 | 17.08 | 20.99 | 27.18 | 16.03 | 43.78 | 19.02 | 24.01 |
| KDA | 43.44 | 31.96 | 35.95 | 36.62 | 47.14 | 23.89 | 65.79 | 53.28 | 56.57 | 73.20 | 49.06 | 18.44 | 23.71 | 28.12 | 15.14 | 41.35 | 20.08 | 24.47 |
| **MDN (Ours)** | **41.62** | **31.51** | 35.60 | 37.43 | 46.93 | 25.17 | 66.43 | 50.28 | 59.25 | 74.30 | **49.42** | 28.07 | 24.65 | 28.01 | 16.95 | 43.01 | 19.89 | **26.76** |
| *1.3B parameters model with 100B training tokens and 1M batch size tokens* | | | | | | | | | | | | | | | | | | |
| Transformer | 17.90 | 18.99 | 52.56 | 51.25 | 58.59 | 27.82 | 71.22 | 58.88 | 61.16 | 82.10 | 57.95 | 51.77 | 46.67 | 39.27 | 26.80 | 57.23 | 21.85 | 40.60 |
| Mamba2 | 18.20 | 19.14 | 52.69 | 49.66 | 58.88 | 29.01 | 71.11 | 54.54 | 60.49 | 80.40 | 57.10 | 25.16 | 35.43 | 35.24 | 22.43 | 53.73 | 22.71 | 32.45 |
| GDN | 16.12 | 18.51 | 52.37 | 50.63 | 58.25 | 28.33 | 72.47 | 56.51 | 58.87 | 82.10 | 57.44 | 27.52 | 33.93 | 33.59 | 24.80 | 55.69 | 21.27 | 32.80 |
| Comba | 15.17 | 18.37 | 53.01 | 51.00 | 59.55 | 29.95 | 72.31 | 56.51 | 62.02 | 83.60 | 58.49 | 35.24 | 38.33 | 35.20 | 25.97 | 55.39 | 21.37 | 35.25 |
| KDA | 16.83 | 19.24 | 53.63 | 52.73 | 59.43 | 30.12 | 71.93 | 57.38 | 59.73 | 83.50 | 58.56 | 28.16 | 32.24 | 33.79 | 24.77 | 54.68 | 22.09 | 32.62 |
| **MDN (Ours)** | **14.87** | **18.03** | 53.50 | 52.97 | 59.55 | 30.52 | 71.87 | 58.48 | 59.63 | 84.00 | **58.82** | 35.42 | 42.74 | 35.67 | 26.09 | 56.54 | 20.36 | **36.14** |

*Table 3.* Performance on LongBench (Bai et al., 2024) tasks with 16K length based on lm-evaluation-harness (Gao et al., 2024).

| Model | Code | | Summarization | | | SingleQA | | | MultiQA | | | Few-shot | | | Avg. |
|---|---|---|---|---|---|---|---|---|---|---|---|---|---|---|---|
| | LCC | RBP | GvR | QMS | MNs | NQA | QQA | MFQ | HQA | 2WM | MUS | TRC | TQA | SSM | |
| Transformer | 38.80 | 11.94 | 5.49 | 7.38 | 17.04 | 0.39 | 3.84 | 7.55 | 0.87 | 3.03 | 0.13 | 9.00 | 8.44 | 3.04 | 8.35 |
| Mamba2 | 38.51 | 30.69 | 5.44 | 16.60 | 14.73 | 2.18 | 5.23 | 13.28 | 5.68 | 7.03 | 2.69 | 32.50 | 31.02 | 4.46 | 15.00 |
| GDN | 42.94 | 30.70 | 7.81 | 15.98 | 16.82 | 2.46 | 5.68 | **13.29** | 5.89 | **9.20** | 3.28 | **55.00** | 34.59 | **26.27** | 19.28 |
| Comba | 44.74 | **36.60** | 9.60 | 16.52 | 16.74 | 2.39 | 6.20 | 13.13 | **6.87** | 7.82 | 3.12 | 48.00 | 35.95 | 5.79 | 18.11 |
| KDA | 37.99 | 33.39 | 10.02 | 15.59 | 16.95 | 2.40 | 5.14 | 12.13 | 6.58 | 8.32 | 3.01 | 37.00 | **46.87** | 25.30 | 18.62 |
| **MDN (Ours)** | **50.50** | **39.13** | **10.24** | **17.20** | **18.85** | **2.63** | 6.27 | 12.21 | 5.81 | 7.26 | **3.63** | 51.00 | 42.13 | 15.65 | **20.18** |

**Needle-In-A-Haystack.** We further evaluate long-range retrieval using the NIAH benchmark from RULER (Hsieh et al., 2024), which evaluates a model's ability to retrieve a specific piece of information (the "needle"). As shown in Figure 5, MDN consistently improves the accuracy across various tasks, especially beyond the training context length. For example, in the challenging multi-needle settings at 8k context length, MDN reaches 38.60 on MK, 35.15 on MQ, and 27.60 on MV, outperforming the strongest baseline on each task by 13.40, 11.45, and 8.95 points, respectively.

### 4.3. Efficiency and Ablation Analysis

**Efficiency Analysis.** We implement MDN in both recurrent and chunkwise parallel forms using Triton. As shown in Figure 6, MDN achieves a decoding latency nearly identical to GDN and Comba, preserving the linear complexity advantage over Transformers. While MDN's training throughput is currently lower than that of Comba and GDN due to its dual-state computation, it attains comparable performance to Mamba2 and KDA by materializing the correction values with a manageable memory overhead. Its competitive decoding speed confirms its practical viability.

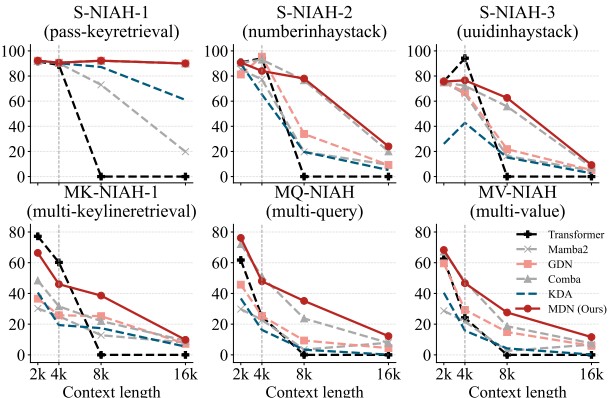

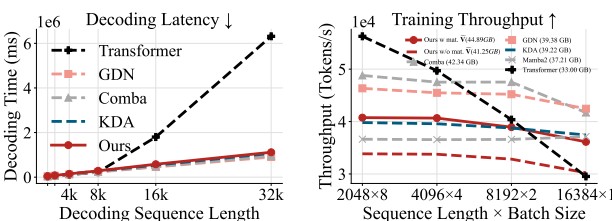

*Figure 6.* (*Left*) Decoding latency and (*Right*) Training throughput comparison of 1.3B models on a single H100 GPU with Triton .

*Figure 5.* The Needle-In-A-Haystack benchmark from RULER (Hsieh et al., 2024) based on lm-evaluation-harness (Gao et al., 2024). The grey vertical line denotes the 4k training length.

**Ablation Study.** Table 4 presents three groups of ablation studies on the 400M model: component-level ablations of MDN, sensitivity analysis of the momentum lower bound, and hybrid variants with full attention. Full table in § H.

*Table 4.* Ablation study on 400M model.

| Model Variant | Lamb. PPL ↓ | Wiki. PPL ↓ | LM ↑ | Retrieval ↑ |
|---|---|---|---|---|
| MDN | **41.62** | **31.51** | **49.42** | **26.76** |
| *Ablation Components* | | | | |
| w/o Output Corr. | 42.31 | 31.72 | 49.19 | 25.52 |
| w/o Momem. | 47.01 | 32.11 | 49.26 | 20.12 |
| w/o Clamp $\mu_{\min}^{\log}$ | NaN due to training divergence | | | |
| w/o $\alpha_{\max}$ | NaN due to training divergence | | | |
| w/o $\beta_{\max}$ | 42.72 | 31.52 | 48.93 | 26.40 |
| $\eta_t = 2\,\mathrm{sigmoid}(\cdot)$ | 49.10 | 31.89 | 49.41 | 25.54 |
| *Sweeping the minimum value of* $\log \mu$ | | | | |
| -2 (reported) | **41.62** | 31.51 | 49.42 | **26.76** |
| -1.5 | 42.65 | 31.50 | 48.94 | 26.03 |
| -1.357 | 44.90 | 31.44 | 49.50 | 25.31 |
| -1 | 43.53 | **31.39** | **49.88** | 25.85 |
| *Hybrid models with Linear Attention: Full Attention* | | | | |
| Mamba2-H (3:1) | 61.27 | 33.73 | 48.34 | 22.24 |
| GDN-H (3:1) | 46.07 | 29.96 | 48.53 | 33.35 |
| Comba-H (3:1) | **40.88** | 29.92 | 49.35 | **34.72** |
| KDA-H (3:1) | 41.78 | **29.49** | 48.93 | 34.46 |
| MDN-H (3:1) | 46.71 | 30.09 | 48.61 | 33.84 |
| MDN-H (7:1) | 42.95 | 30.32 | **49.68** | 34.37 |

For the component-level ablations, MDN still outperforms GDN and Comba after removing the output correction, suggesting that momentum alone provides substantial gains. The stability-aware parameterization is also important for stable training: removing the lower bound on $\log \mu$ or the constraint on $\alpha_{\max}$ leads to divergence. In addition, weakening the constraint on $\beta_{\max}$ or changing the activation function of $\eta_t = 2\,\mathrm{sigmoid}(\cdot)$ degrades performance.

For the $\mu_{\min}^{\log}$ sensitivity analysis, the reported setting $\mu_{\min}^{\log} = -2$ provides the best overall trade-off, achieving the lowest LAMBADA perplexity and the highest retrieval accuracy. Larger lower bounds remain trainable but generally weaken retrieval performance, suggesting that overly restricting the momentum range may limit expressivity.

For the hybrid variants, the 3:1 linear/full-attention ratio is widely adopted in recent hybrid architectures, such as Kimi (Team et al., 2025) and Qwen-3.5 (Team, 2026). When full attention is used more sparsely with a 7:1 ratio, MDN-H further improves the LM average while maintaining competitive retrieval accuracy. This suggests that MDN can reduce the dependence on full-attention layers, making it a promising building block for more efficient hybrid architectures.

**Hidden State Statistics Analysis.** Inspired by Buitrago & Gu (2025); Dohare et al. (2024), we analyze fast weight dynamics by measuring the average change norm $\Delta\mathbf{S}_t = \frac{1}{n}\sum_{i=1}^{n}\|\mathbf{S}_t - \mathbf{S}_{t-1}\|_F$ during recurrent decoding. This metric captures the magnitude of fast weight updates between adjacent decoding steps, providing a direct view of how much the recurrent memory changes over time. As shown in Figure 7, MDN exhibits consistently larger change norms than Comba and GDN across most decoding steps. This indicates that MDN updates its fast weight state more actively

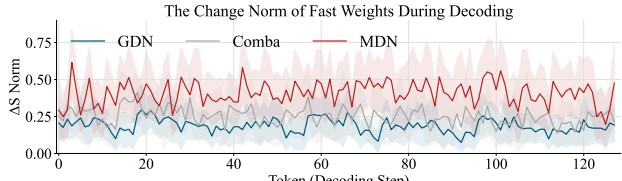

*Figure 7.* The change norm of fast weights during decoding.

during decoding, whereas the Comba and GDN show relatively smaller state variations. Such stronger state variation is consistent with the empirical improvements observed in retrieval and downstream evaluations, suggesting that richer dynamics may be beneficial for sequence modeling.

## 5. Conclusion

In this paper, we present Momentum DeltaNet, which scales stepwise momentum in linear attention through an efficient chunkwise parallel algorithm. By resolving computational bottlenecks and integrating constrained gating mechanisms, our model achieves a superior balance between expressive dynamics and efficiency. Experimental results confirm that Momentum DeltaNet outperforms existing linear attention baselines across a range of downstream tasks. In future work, we will explore sophisticated gating strategies and further optimize kernels to maximize hardware efficiency.

## Acknowledgements

This work is supported in part by the Guangdong Basic and Applied Basic Research Foundation (No. 2025A1515011758), the Youth S&T Talent Support Programme of Guangdong Provincial Association for Science and Technology (SKXRC2025460) and Guangdong Provincial Key Lab of Integrated Communication, Sensing and Computation for Ubiquitous Internet of Things (No.2023B1212010007).

We would like to express our sincere gratitude to the flash-linear-attention community for their insightful discussions and open-source framework, which were helpful during the development of this work. We are also grateful to the Scientific Spaces blog for its insightful mathematical discussions.

## Impact Statement

This paper presents work whose goal is to advance the field of machine learning. Our contribution is primarily methodological, focusing on efficient sequence modeling and linear attention for large language models. All experiments are conducted on public academic benchmarks, and we do not introduce new datasets involving personal or sensitive information. The broader societal impacts of this work are mainly those associated with advances in large language models and efficient machine learning systems in general, none of which we feel must be specifically highlighted here.

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

## A. Notation

Following Yang et al. (2024a); Team et al. (2025), we use bold upper-case letters for matrices (e.g., $\mathbf{S}$, $\mathbf{Q}$), bold lower-case letters for *column* vectors (e.g., $\boldsymbol{q}_t$, $\boldsymbol{k}_t$) and italic upper-case letters for learnable parameter matrices (e.g., $\boldsymbol{W}_k$). We denote the $t$-th row vector of a matrix $\mathbf{Q}$ as $\boldsymbol{q}_t^\top$, where $\top$ denotes transposition. $\mathbf{M}$ and $\mathbf{M}^-$ denote lower-triangular masks with and without diagonal elements, respectively.

**Chunkwise Formulation** Consider length sequence of $L$ split into $L/C$ chunks, each of chunk length $C$. We define $\square_{[t]} \in \mathbb{R}^{C \times d}$ for $\square \in \{\mathbf{Q}, \mathbf{K}, \mathbf{V}, \cdots\}$, where $\square_{[t]}$ stacks the vectors within the $t$-th chunk. The $r$-th element of the $t$-th chunk is $\square_{[t]}^r = \square_{tC+r}$. Here, $t \in [0, L/C)$ and $r \in [1, C]$. State matrices are re-indexed such that $\mathbf{S}_{[t]}^i = \mathbf{S}_{tC+i}$, with $\mathbf{S}_{[t]} := \mathbf{S}_{[t]}^0 = \mathbf{S}_{[t-1]}^C$, that is the initial state of a chunk is the last state of the previous chunk.

**Decay Formulation** We use the bar symbol to denote the cumulative product $\bar{x}_r := \prod_{i=1}^r x_i$. Consequently, for $i \geq j$, $\prod_{k=j+1}^i x_k = \bar{x}_i / \bar{x}_j$. For chunkwise formulation, it can extend to $\bar{x}_{[t]}^r = \prod_{i=tC+1}^{tC+r} x_i$ for $k \in [1, C]$. For $i \geq j$ with $i, j \in [1, C]$, then $\prod_{k=j+1}^i x_{[t]}^k = \bar{x}_{[t]}^i / \bar{x}_{[t]}^j$. We define the chunk vector $\bar{\boldsymbol{\alpha}}_{[t]}^{j \to i} \in \mathbb{R}^C$ as ordered stack from scalar $\bar{\alpha}_{[t]}^j$ to $\bar{\alpha}_{[t]}^i$, we abbreviate $\bar{\boldsymbol{\alpha}}_{[t]}$ when $j = 1, i = C$.

## B. Extended Related Work

**Linear Attention with Gating.** The $O(N^2)$ complexity of standard self-attention (Vaswani et al., 2017) has spurred the development of linear-time alternatives. The general formulation of Linear Attention with Gating can be unified as Eq. (20). The $\odot$ denotes the element-wise Hadamard product. The $\mathbf{S}_t$ denotes the accumulated memory state, and $\mathbf{G}_t$ acts as memory gating. In early vanilla Linear Transformers, the gating is identity $\mathbf{G}_t = \mathbf{I}$ (Katharopoulos et al., 2020; Choromanski et al., 2020), which leads to an unbounded summation where early tokens can dominate and saturate the state. To resolve this, the field moved toward data-independent decay, such as RetNet (Sun et al., 2023), which employs a fixed exponential decay $\mathbf{G}_t = \alpha \mathbf{I} \in (0, 1)$. Modern architectures like Mamba (Gu & Dao, 2023; Dao & Gu, 2024) introduce Selective Gating ($\mathbf{G}_t = \alpha_t \mathbf{I}$), where the decay becomes a function of the input, enabling the model to selectively forget irrelevant information to regulate the recurrent state. Gated Linear Attention (GLA) (Yang et al., 2024a) further extends the scalar decay to vector decay as $\mathbf{G}_t = \text{Diag}(\boldsymbol{\alpha}_t)$, introducing fine-grained, channel-wise control over the forgetting process. The input gate constraint according to memory gating to balance stability and capacity further refine the gated linear paradigm, like the MetaLA (Chou et al., 2024) HGRN1&2 (Qin et al., 2023; 2024b).

$$\mathbf{S}_t = \mathbf{G}_t \odot \mathbf{S}_{t-1} + \beta_t \boldsymbol{k}_t \boldsymbol{v}_t^\top \quad \text{(Memory Gating)} \tag{20}$$

**Linear Attention with Correction.** Standard linear attention follows a Hebbian update rule, in which new associations ($\boldsymbol{k}_t \boldsymbol{v}_t^\top$) are passively superimposed onto the existing state. While efficient, this form of accumulation is prone to capacity saturation and sensitivity to input noise. The correction framework instead interprets the recurrent state as a form of fast weight memory, optimized through online learning. From this perspective, the hidden state functions as a Fast Weight Programmer (FWP) (Schmidhuber, 1992; Schlag et al., 2021), updated via online gradient descent. This gradient descent can be tranform to the value correction form, $\widetilde{\boldsymbol{v}}_t = \boldsymbol{v}_t - \mathbf{S}_{t-1}^\top \boldsymbol{k}_t$, which represents the reconstruction error of the current value (Eq. (21), left). Updating the state along this error direction effectively corrects previous associations and implicitly orthogonalizes the memory when $\mathbf{G}_t = \mathbf{I}$, leading to more efficient utilization of limited capacity (Schlag et al., 2021).

Gated DeltaNet (GDN) (Yang et al., 2025) combines the Delta rule with input-dependent decay by setting $\mathbf{G}_t = \alpha_t \mathbf{I}$ and modifying the correction term to $\widetilde{\boldsymbol{v}}_t = \boldsymbol{v}_t - \alpha_t \mathbf{S}_{t-1}^\top \boldsymbol{k}_t$, thereby jointly controlling forgetting and correction. More recently, Kimi Linear Attention (Kimi Delta Attention, KDA) (Team et al., 2025) extends this framework by introducing channel-wise diagonal gating, $\mathbf{G}_t = \text{Diag}(\boldsymbol{\alpha}_t)$, together with a correspondingly gated correction term $\widetilde{\boldsymbol{v}}_t = \boldsymbol{v}_t - \text{Diag}(\boldsymbol{\alpha}_t) \mathbf{S}_{t-1}^\top \boldsymbol{k}_t$. Building upon the delta rule, Comba (Hu et al., 2025) further introduces query correction as $\widetilde{\boldsymbol{q}}_t = \boldsymbol{q}_t - d\boldsymbol{k}_t$, which has been shown to yield additional performance gains. More variant delta rule has also been explored, like RWKV7 (Peng et al., 2025). Recent work on Longhorn (Liu et al., 2024) derives an adaptive learning rate by regarding recurrent updates as the closed-form solution to an online learning objective. Error-Free Linear Attention (EFLA) (Lei et al., 2025) formulates linear attention as the exact solution of a continuous-time dynamical system. An alternative correction strategy operates on keys rather than values, by setting $\widetilde{\boldsymbol{k}}_t = \boldsymbol{k}_t - \mathbf{S}_{t-1} \boldsymbol{v}_t$, which yields an update equivalent to Oja's rule (Oja, 1989) (Eq. (21), right). However, whether Oja's rule can scale to large-scale linear attention remains an open question.

$$\mathbf{S}_t = \mathbf{G}_t \odot \mathbf{S}_{t-1} + \beta_t \boldsymbol{k}_t \widetilde{\boldsymbol{v}}_t^\top \quad \text{(Value Correction)}, \qquad \mathbf{S}_t = \mathbf{G}_t \odot \mathbf{S}_{t-1} + \beta_t \widetilde{\boldsymbol{k}}_t \boldsymbol{v}_t^\top \quad \text{(Key Correction)}. \tag{21}$$

**Linear Attention with Expanding Memory.** Beyond single-state gating or error correction, recent work expands memory dynamics by composing multiple gated or delta-based recurrences. ABC (Peng et al., 2022) couples two vanilla linear attention modules, while GSA (Zhang et al., 2024) further introduces input-dependent gating. MesaNet (von Oswald et al., 2025), derived from an in-context regression objective, solves a test-time linear loss via conjugate gradient and can be viewed as a dual recurrence of GLA. Log Linear Attention (Guo et al., 2025) scales memory in models such as Mamba2 and Gated DeltaNet using a Fenwick tree, achieving $O(\log N)$ recurrent capacity with subquadratic cost.

**State Space Models.** The State Space Models (SSMs) became very State Space Models (SSMs) gained prominence prior to linear attention due to their linear complexity in context length and strong interpretability. Early representative models include the Structured State Space Sequence model (S4) (Gu et al., 2022), followed by Diagonal State Space (DSS) (Gupta et al., 2022), Gated State Space (GSS) models (Mehta et al., 2023), S5 (Smith et al., 2023), Bidirectional Gated SSM (BiGS) (Wang et al., 2022), H3 (Fu et al., 2023), and the more recent Mamba family (Gu & Dao, 2023; Dao & Gu, 2024; Lahoti et al., 2026).

**Modern Non-Linear RNN** Modern non-linear recurrent neural networks (RNNs) revisit classical architectures such as LSTM (Hochreiter & Schmidhuber, 1997) and GRU (Chung et al., 2014). xLSTM (Beck et al., 2024) alleviates the scalar memory bottleneck via exponential gating and a Matrix Long Short-Term Memory (mLSTM) block, which employs a fully parallelizable covariance-style update to increase memory capacity. In parallel, the Hawk and Griffin models (De et al., 2024) introduce the Real-Gated Linear Recurrent Unit (RG-LRU), combining gated recurrence with a highly optimized diagonal structure for stable non-linear dynamics at scale.

**Test Time Optimization Perspective.** Pushing the memory paradigm further, Test-Time Training (TTT) (Sun et al., 2024) redefines the hidden state as the weights of an inner model that are updated online via regression at inference time, casting sequence modeling as a continual optimization process. Building on this view, Titans (Behrouz et al., 2025b) augment batched gradient descent with momentum, while subsequent work unifies a broad class of efficient foundation models from a test time regression perspective (Wang et al., 2025b). Extending Titans, Atlas (Behrouz et al., 2025a) adopts a sliding-window formulation closely related to the Mesa layer (Von Oswald et al., 2023). More broadly, these approaches connect to earlier test time optimization methods (Krause et al., 2018; Clark et al., 2022; Liu et al., 2026a). From an optimization perspective, linear attention and these methods can be viewed within a unified framework, as illustrated in Figure 8.

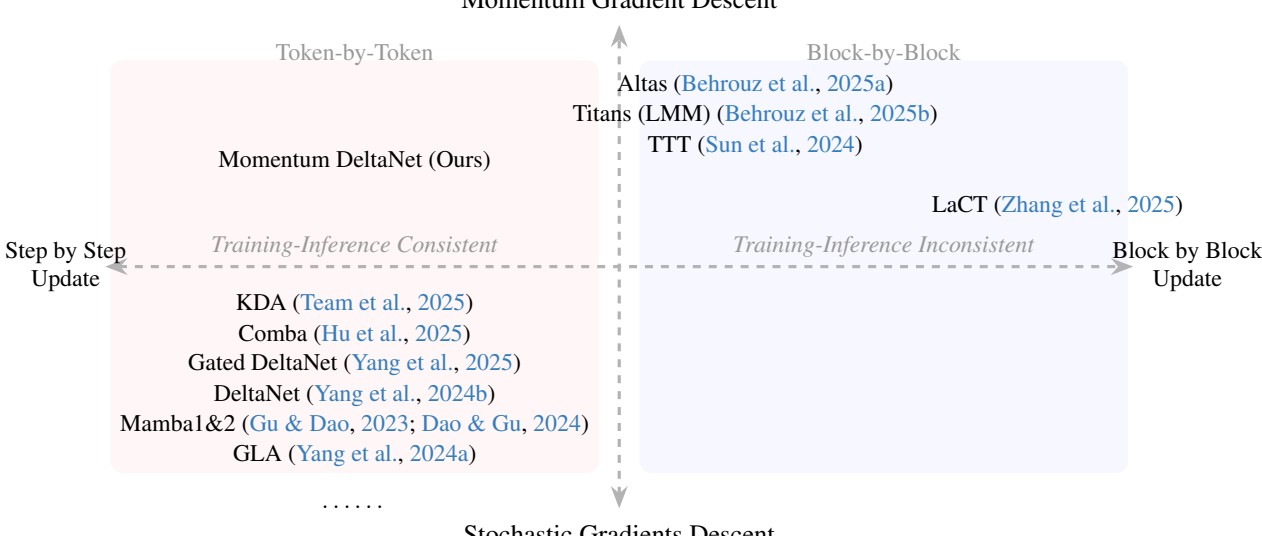

*Figure 8.* Optimizer Perspective for Auto-Regression Sequence Modeling.

## C. Chunkwise Parallel Derivation for Momentum Delta Rule.

In this section, we derive a general parallel formulation of the momentum delta rule, which naturally extends to a chunkwise parallel implementation. We begin by recalling the recurrent momentum formulation in Eqs. (4) and (5):

$$
\begin{aligned}
\mathbf{M}_t &= \mu_t \mathbf{M}_{t-1} - \eta_t \boldsymbol{k}_t \widetilde{\boldsymbol{v}}_t^\top, \\
\mathbf{S}_t &= \alpha_t \mathbf{S}_{t-1} - \beta_t \mathbf{M}_t,
\end{aligned}
\tag{22}
$$

where $\mathbf{S}_t, \mathbf{M}_t \in \mathbb{R}^{d_k \times d_v}$, $\boldsymbol{k}_t \in \mathbb{R}^{d_k}$, $\boldsymbol{v}_t \in \mathbb{R}^{d_v}$, and $\alpha_t, \beta_t, \mu_t, \eta_t$ are scalars.

The correction term is defined as $\widetilde{\boldsymbol{v}}_t := \boldsymbol{v}_t - \mathbf{S}_{t-1}^\top \boldsymbol{p}_t$ which aligns the formulation with Comba (Hu et al., 2025). Following Gated DeltaNet (Yang et al., 2025), we set $\boldsymbol{p}_t = \alpha_t \boldsymbol{k}_t$. From the test-time training perspective, $\boldsymbol{k}_t$ can be viewed as the input to a fast weight memory $\mathbf{S}_t$, while $\widetilde{\boldsymbol{v}}_t$ corresponds to the output errors.

We now consider the momentum case with $\mu_t \neq 0$ and expand the recurrent updates. With the expansion of the recurrence,

$$
\mathbf{M}_t = \bar{\mu}_t \mathbf{M}_0 - \sum_{i=1}^{t} \frac{\bar{\mu}_t}{\bar{\mu}_i} \boldsymbol{k}_i \widetilde{\boldsymbol{v}}_i^\top
\tag{23}
$$

$$
\mathbf{S}_t = \bar{\alpha}_t \mathbf{S}_0 - \sum_{i=1}^{t} \beta_i \frac{\bar{\alpha}_t}{\bar{\alpha}_i} \mathbf{M}_i
\tag{24}
$$

$$
= \bar{\alpha}_t \mathbf{S}_0 - \sum_{i=1}^{t} \beta_i \frac{\bar{\alpha}_t}{\bar{\alpha}_i} \left( \bar{\mu}_i \mathbf{M}_0 - \sum_{j=1}^{i} \frac{\bar{\mu}_i}{\bar{\mu}_j} \boldsymbol{k}_j \widetilde{\boldsymbol{v}}_j^\top \right)
\tag{25}
$$

$$
= \bar{\alpha}_t \mathbf{S}_0 - \sum_{i=1}^{t} \beta_i \frac{\bar{\alpha}_t}{\bar{\alpha}_i} \bar{\mu}_i \mathbf{M}_0 + \sum_{i=1}^{t} \beta_i \frac{\bar{\alpha}_t}{\bar{\alpha}_i} \sum_{j=1}^{i} \frac{\bar{\mu}_i}{\bar{\mu}_j} \boldsymbol{k}_j \widetilde{\boldsymbol{v}}_j^\top
\tag{26}
$$

$$
= \bar{\alpha}_t \mathbf{S}_0 - \bar{\alpha}_t \left( \sum_{i=1}^{t} \frac{\beta_i \bar{\mu}_i}{\bar{\alpha}_i} \right) \mathbf{M}_0 + \bar{\alpha}_t \sum_{i=1}^{t} \bar{\mu}_i^{-1} \left( \sum_{j=i}^{t} \frac{\beta_j \bar{\mu}_j}{\bar{\alpha}_j} \right) \boldsymbol{k}_i \widetilde{\boldsymbol{v}}_i^\top
\tag{27}
$$

The key transformation from Eq.(26) to Eq.(27) is according to the:

$$
\sum_{i=1}^{t} \sum_{j=1}^{i} a_i \cdot b_j = \sum_{j=1}^{t} \sum_{i=j}^{t} a_i \cdot b_j = \sum_{i=1}^{t} \sum_{j=i}^{t} a_j \cdot b_i.
\tag{28}
$$

Geometrically, this reordering corresponds to exchanging the order of accumulation over the lower-triangular region $\{(i,j) \mid 1 \leq j \leq i \leq t\}$ in the $(i,j)$ index plane. Both summations traverse the same triangular domain, but along orthogonal directions: the original form aggregates contributions row-wise, while the reordered form aggregates column-wise. This geometric reinterpretation is crucial for parallelization, as it exposes the dependency structure explicitly and allows prefix style aggregation within each column, which can be computed efficiently in parallel. Then the final momentum and fast weight can be computed according to the parallel form as shown in Eq. (29) and (30),

$$
\mathbf{M}_t = \bar{\mu}_t \mathbf{M}_0 - \underbrace{\sum_{i=1}^{t} \frac{\bar{\mu}_t}{\bar{\mu}_i} \boldsymbol{k}_i \widetilde{\boldsymbol{v}}_i^\top}_{\text{Recurrent Form}} = \bar{\mu}_t \mathbf{M}_0 - \underbrace{\left( \mathrm{Diag}\left( \frac{\bar{\mu}_t}{\bar{\mu}} \right) \cdot \mathbf{K} \right)^\top \widetilde{\mathbf{V}}}_{\text{Parallel Form}}
\tag{29}
$$

$$
\mathbf{S}_t := \underbrace{\bar{\alpha}_t \mathbf{S}_0 - b_t \mathbf{M}_0 + \sum_{i=1}^{t} \gamma_{t,i} \boldsymbol{k}_i \widetilde{\boldsymbol{v}}_i^\top}_{\text{Recurrent Form}} = \bar{\alpha}_t \mathbf{S}_0 - b_t \mathbf{M}_0 + \underbrace{\left( \mathrm{Diag}\left( \boldsymbol{\Gamma}_{t,:} \right) \cdot \mathbf{K} \right)^\top \widetilde{\mathbf{V}}}_{\text{Parallel Form}}
\tag{30}
$$

The corresponding coefficient in Eq. (27), Eq. (29) and Eq. (30) as defined below,

$$
\bar{\mu}_t := \prod_{j=1}^{t} \mu_j \quad \bar{\alpha}_t := \prod_{j=1}^{t} \alpha_j \quad c_t := \sum_{i=1}^{t} \frac{\beta_i \bar{\mu}_i}{\bar{\alpha}_i} \quad b_t := \bar{\alpha}_t c_t \quad \gamma_{t,i} := \frac{\bar{\alpha}_t}{\bar{\mu}_i} \cdot (c_t - c_{i-1})
\tag{31}
$$

where $\mathbf{\Gamma}_{t,:}$ in Eq. (30) is the last row of the lower triangle matrix $\mathbf{\Gamma}$, with $\mathbf{\Gamma}_{(i,j)} = \gamma_{i,j}$ for $i \geq j$ else equal 0. As defined before, we need to solve this correction value $\widetilde{v}_t$ transform the recursion form as iteration form, recall the definition before $\widetilde{v}_t := v_t - \mathbf{S}_{t-1}^\top p_t$, We substitute Eq. (30) into this definition,

$$\widetilde{v}_t^\top := v_t^\top - p_t^\top \mathbf{S}_{t-1} \tag{32}$$

$$\widetilde{v}_t^\top = v_t^\top - p_t^\top \left( \bar{\alpha}_{t-1} \mathbf{S}_0 - b_{t-1} \mathbf{M}_0 + \sum_{i=1}^{t-1} \gamma_{t-1,i}\, k_i \widetilde{v}_i^\top \right) \tag{33}$$

$$= v_t^\top - \bar{\alpha}_{t-1} p_t^\top \mathbf{S}_0 + b_{t-1} p_t^\top \mathbf{M}_0 - p_t^\top \sum_{i=1}^{t-1} \gamma_{t-1,i}\, k_i \widetilde{v}_i^\top \tag{34}$$

$$\left( 1 + p_t^\top \sum_{i=1}^{t-1} \gamma_{t-1,i} k_i \right) \cdot \widetilde{v}_i^\top = v_t^\top - \bar{\alpha}_{t-1} p_t^\top \mathbf{S}_0 + b_{t-1} p_t^\top \mathbf{M}_0 \tag{35}$$

$$\left( \mathbf{I} + \mathbf{\Gamma}^- \odot \mathbf{P}\mathbf{K}^\top \right) \cdot \widetilde{\mathbf{V}} = \mathbf{V} - (\mathrm{Diag}(\bar{\alpha}^-)\mathbf{P}\mathbf{S}_0 + \mathrm{Diag}(b^-)\mathbf{P}\mathbf{M}_0 \tag{36}$$

where $\mathbf{\Gamma}^- \in \mathbb{R}^{L \times L}$ is a strict lower triangle matrix . The correction term $\widetilde{v}_t$ can now be solved in parallel by organizing Eq. (36). Collecting terms yields

$$\widetilde{\mathbf{V}} = \mathbf{U} - \mathbf{Y} \cdot \mathbf{S}_0 + \mathbf{Z} \cdot \mathbf{M}_0 \tag{37}$$

$$\mathbf{U} = \mathbf{T} \cdot \mathbf{V} \qquad \mathbf{Y} = \mathbf{T} \cdot (\mathrm{Diag}(\bar{\alpha}^-)\mathbf{P}) \qquad \mathbf{Z} = \mathbf{T} \cdot (\mathrm{Diag}(b^-)\mathbf{P}) \tag{38}$$

$$\mathbf{T} = \mathrm{Tril}\left( \mathbf{I} + \left( \mathbf{P}\mathbf{K}^\top \odot \mathbf{\Gamma}^- \right) \right)^{-1} \tag{39}$$

Although this full parallel formulation is computationally inefficient, it serves as the foundation for an efficient chunkwise-parallel implementation. Each chunk is algebraically equivalent to the global parallel form. By computing the coefficients $\bar{\alpha}_{[t]}^{\log}, \bar{\mu}_{[t]}^{\log}, \beta_{[t]}, c_{[t]} \in \mathbb{R}^C$ and $\mathbf{\Gamma}_{[t]} \in \mathbb{R}^{C \times C}$ in Eq. (31) within each chunk (details in Appendix D), the updates can be decomposed into inter-chunk and intra-chunk components.

Specifically, for chunk $[t]$, the output is computed as

$$\mathbf{O}_{[t]} = \underbrace{\mathrm{Diag}(\bar{\alpha}_{[t]}) \cdot \mathbf{Q}_{[t]}\mathbf{S}_{[t]} - \mathrm{Diag}(b_{[t]}) \cdot \mathbf{Q}_{[t]}\mathbf{M}_{[t]}}_{\text{Inter-Chunk}} + \underbrace{\left( (\mathbf{Q}_{[t]}\mathbf{K}_{[t]}^\top) \odot \mathbf{\Gamma}_{[t]} \right) \cdot \widetilde{\mathbf{V}}_{[t]}}_{\text{Intra-Chunk}} \qquad \in \mathbb{R}^{C \times d_v} \tag{40}$$

The hidden states are then updated as

$$\mathbf{M}_{[t+1]} = \bar{\mu}_{[t]}^C \cdot \mathbf{M}_{[t]} - \left( \mathrm{Diag}\left( \frac{\bar{\mu}_{[t]}^C}{\bar{\mu}_{[t]}} \right) \cdot \mathbf{K}_{[t]} \right)^\top \widetilde{\mathbf{V}}_{[t]} \tag{41}$$

$$\mathbf{S}_{[t+1]} = \bar{\alpha}_{[t]}^C \cdot \mathbf{S}_{[t]} - b_{[t]}^C \cdot \mathbf{M}_{[t]} + \left( \mathrm{Diag}\left( \mathbf{\Gamma}_{[t]}^C \right) \cdot \mathbf{K}_{[t]} \right)^\top \widetilde{\mathbf{V}}_{[t]}, \tag{42}$$

where $\mathbf{\Gamma}_{[t]}^C$ is the $C$-th row (last row) vector of the $t$-th chunk of causal mask $\mathbf{\Gamma}_{[t]} \in \mathbb{R}^{C \times C}$. And the correlation value $\widetilde{\mathbf{V}}_{[t]}$,

$$\widetilde{\mathbf{V}}_{[t]} = \mathbf{U}_{[t]} - \mathbf{Y}_{[t]}\mathbf{S}_{[t]} + \mathbf{Z}_{[t]}\mathbf{M}_{[t]} \qquad \in \mathbb{R}^{C \times d_v}, \tag{43}$$

where $\mathbf{U}_{[t]} \in \mathbb{R}^{C \times d_v}$ and $\mathbf{Y}_{[t]}, \mathbf{Z}_{[t]} \in \mathbb{R}^{C \times d_k}$ are computed by $\mathbf{T}_{[t]} \in \mathbb{R}^{C \times C}$, the detailed computation are blow,

$$\mathbf{U}_{[t]} = \mathbf{T}_{[t]} \cdot \mathbf{V}_{[t]}, \tag{44}$$

$$\mathbf{Y}_{[t]} = \mathbf{T}_{[t]} \cdot \left( \mathrm{Diag}(\bar{\alpha}_{[t]}^{0 \to C-1}) \cdot \mathbf{P}_{[t]} \right) \tag{45}$$

$$\mathbf{Z}_{[t]} = \mathbf{T}_{[t]} \cdot \left( \mathrm{Diag}(b_{[t]}^{0 \to C-1}) \cdot \mathbf{P}_{[t]} \right) \tag{46}$$

$$\text{where} \quad \mathbf{T}_{[t]} = \mathrm{Tril}\left( \mathbf{I}_{[t]} + \left( \mathbf{P}_{[t]}\mathbf{K}_{[t]}^\top \odot \mathbf{\Gamma}_{[t]}^- \right) \right)^{-1} \tag{47}$$

Beyond the scope of MDN, the proposed geometric decoupling strategy offers a principled perspective for alleviating dependency bottlenecks in certain higher-order linear dynamical systems. It provides a reusable mathematical template for parallelizing a class of non-stationary linear recurrences with structured dependencies. Moreover, this formulation suggests the potential for extending linear attention mechanisms to incorporate more advanced optimization-inspired update rules (e.g., Nesterov Momentum, Adam and Muon scaling) into the linear attention paradigm.

## D. Coefficients Chunkwise Parallelization

We first recall the definitions of the accumulated coefficients from Eq. (9):

$$\bar{\mu}_t := \prod_{j=1}^{t} \mu_j, \quad \bar{\alpha}_t := \prod_{j=1}^{t} \alpha_j, \quad c_t := \sum_{i=1}^{t} \frac{\beta_i \bar{\mu}_i}{\bar{\alpha}_i}, \qquad b_t := \bar{\alpha}_t c_t, \qquad \gamma_{t,i} := \frac{\bar{\alpha}_t}{\bar{\mu}_i}(c_t - c_{i-1}) \quad \text{for} \quad i <= t. \tag{48}$$

The above coefficients can be computed chunkwise in parallel in the log-domain. For each chunk indexed by [t], we compute

$$\bar{\boldsymbol{\mu}}_{[t]}^{\log} = \text{cumsum}(\boldsymbol{\mu}_{[t]}^{\log}), \quad \bar{\boldsymbol{\alpha}}_{[t]}^{\log} = \text{cumsum}(\boldsymbol{\alpha}_{[t]}^{\log}) \qquad \in \mathbb{R}^C, \tag{49}$$

$$\boldsymbol{c}_{[t]}^{\log} = \log(\text{cumsum}(\exp(\bar{\boldsymbol{\mu}}_{[t]}^{\log} - \bar{\boldsymbol{\alpha}}_{[t]}^{\log} + \boldsymbol{\beta}_{[t]}^{\log}))) \qquad \in \mathbb{R}^C, \tag{50}$$

$$\boldsymbol{b}_{[t]} = \exp(\boldsymbol{\mu}_{[t]}^{\log} + \boldsymbol{c}_{[t]}^{\log}) \qquad \in \mathbb{R}^C, \tag{51}$$

$$\boldsymbol{\Gamma}_{[t]} = \exp(\mathbf{A}_{[t]}^{\log} + \mathbf{C}_{[t]}^{\log}) \qquad \in \mathbb{R}^{C \times C}. \tag{52}$$

Here, $\boldsymbol{\Gamma}[t]$ is a lower-triangular matrix. The log-domain matrix $\mathbf{A}^{\log}[t]$ is defined as

$$(\mathbf{A}_{[t]}^{\log})_{ij} = (\bar{\boldsymbol{\alpha}}_{[t]}^{\log})_i - (\bar{\boldsymbol{\mu}}_{[t]}^{\log})_j, \quad i \geq j \in [1, C], \tag{53}$$

We further compute $\mathbf{C}^{\log}[t]$ in the log-domain as:

$$(\mathbf{C}_{[t]}^{\log})_{ij} = \log\left(\exp(\boldsymbol{c}_{[t]}^{\log})_i - \exp(\boldsymbol{c}_{[t]}^{\log})_{j-1}\right)$$

$$= \log\left(\exp(\boldsymbol{c}_{[t]}^{\log})_i \left(1 - \frac{\exp(\boldsymbol{c}_{[t]}^{\log})_{j-1}}{\exp(\boldsymbol{c}_{[t]}^{\log})_i}\right)\right)$$

$$= (\boldsymbol{c}_{[t]}^{\log})_i + \log\left(1 - \exp(\boldsymbol{s}_{[t]}^{\log})_{ij}\right)$$

where $(\boldsymbol{s}_{[t]}^{\log})_{ij} = (\boldsymbol{c}_{[t]}^{\log})_{j-1} - (\boldsymbol{c}_{[t]}^{\log})_i \in (0, 1]$ with $i \geq j$. To avoid numerical issues when $(\boldsymbol{s}^{\log}[t])ij = 0$ (which would lead to $\log 0^+ = -\infty$), we rewrite Eq. (52) as,

$$\boldsymbol{\Gamma}_{[t]} = \exp(\widetilde{\mathbf{A}}_{[t]}^{\log}) \odot \left(1 - \exp(\mathbf{S}_{[t]}^{\log})\right), \tag{54}$$

where the corrected log-coefficient is given by $(\widetilde{\mathbf{A}}_{[t]}^{\log})_{ij} = (\bar{\boldsymbol{\alpha}}_{[t]}^{\log} + \boldsymbol{c}_{[t]}^{\log})_i - (\bar{\boldsymbol{\mu}}_{[t]}^{\log})_j$. The operator cumsum denotes a prefix-sum computation within each chunk, which can be implemented with $O(\log C)$ parallel complexity. In practice, we implement the log-cumsum-exp operation in Triton by broadcasting the vectors to a masked lower-triangular matrix, subtracting the per-row maximum for numerical stability, and then accumulating in the log-domain, as detailed in Algorithm 1.

---

**Algorithm 1** Triton-like Pseudo Code for Chunkwise log-sum-exp for computing $\log \mathbf{c}_t$

---

**Require:** $\log \mathbf{a} \in \mathbb{R}^C$, $\log \mathbf{m} \in \mathbb{R}^C$, $\boldsymbol{\beta} \in (0,1)^C$, $\epsilon > 0$, Chunk index $i_t$, Chunk size $C \in \{16, 32, 64\}$
**Ensure:** $\log \mathbf{c}_t \in \mathbb{R}^C$
1: $\log \boldsymbol{\beta} \leftarrow \log(\boldsymbol{\beta} + \epsilon)$
2: $\log \mathbf{c} \leftarrow \log \boldsymbol{\beta} + \text{cumsum}(\log \mathbf{m} - \log \mathbf{a}, \text{axis} = 0)$
3: $\mathbf{o} \leftarrow i_t \cdot C + \text{arange}(0, C)$      // $\mathbf{o} \in \mathbb{R}^C$
4: $\mathbf{L} \leftarrow \text{where}(\mathbf{o}[:, \text{None}] \geq \mathbf{o}[\text{None}, :], \log \mathbf{c}[\text{None}, :], -\infty)$      // $\mathbf{L} \in \mathbb{R}^{C \times C}$
5: $\mathbf{r} \leftarrow \max(\mathbf{L}, \text{axis} = 1)$      // $\mathbf{r} \in \mathbb{R}^C$
6: $\mathbf{s} \leftarrow \text{sum}(\exp(\mathbf{L} - \mathbf{r}[:, \text{None}]), \text{axis} = 1)$      // $\mathbf{s} \in \mathbb{R}^C$
7: $\log \mathbf{c}_t \leftarrow \log(\mathbf{s}) + \mathbf{r}$

---

## E. Pytorch-like Pseudo Code for Recurrent and Chunkwise MDN

```
1  def recurrent_momentum_delta_rule(
2          q: torch.Tensor,
3          k: torch.Tensor,
4          v: torch.Tensor,
5          p: torch.Tensor,      # we use p = α · k out of this function
6          log_alpha: torch.Tensor,
7          log_mu: torch.Tensor,
8          beta: torch.Tensor,
9          eta: torch.Tensor,
10         scale: float = None,
11         initial_S: torch.Tensor = None,
12         initial_M: torch.Tensor = None,
13         output_final_state: bool = False,
14 ):
15     q, k, v, p, log_alpha, log_mu, beta, eta = map(lambda x: x.to(torch.float32),
16                                                     [q, k, v, p, log_alpha, log_mu, beta,
       eta]
17                                                     )
18     B, T, H, DK, DV = *k.shape, v.shape[-1]
19
20     if scale is None:
21         scale = 1 / (q.shape[-1] ** 0.5)
22
23     q = q * scale
24     S_prev = torch.zeros(B, H, DK, DV).to(v)
25     M_prev = torch.zeros(B, H, DK, DV).to(v)
26
27     if initial_M is not None: M_prev = initial_M
28     if initial_S is not None: S_prev = initial_S
29
30     out = torch.zeros_like(v)
31     for i in range(T):
32         k_t = k[:, i]   # B, H, DK
33         q_t = q[:, i]   # B, H, DK
34         v_t = v[:, i]   # B, H, DV
35         p_t = p[:, i]   # B, H, DK
36
37         mu_i = log_mu[:, i].exp().view(B, H, 1, 1)        # B, H, 1, 1
38         beta_i = beta[:, i].view(B, H, 1, 1)              # B, H, 1, 1
39         alpha_i = log_alpha[:, i].exp().view(B, H, 1, 1)  # B, H, 1, 1
40         eta_i = eta[:, i].unsqueeze(-1)  # B, H, 1
41
42         # Delta Grad
43         w_t = - (v_t.unsqueeze(-2) - p_t.unsqueeze(-2) @ S_prev)
44         # Mt: Momentum, St: Fast weight
45         # (B, H, k, 1)  @  (B, H, 1, V) = (B, H, K, V)
46         Mt = mu_i * M_prev + (eta_i * k_t).unsqueeze(-1) @ w_t
47         St = alpha_i * S_prev - beta_i * Mt
48
49         # (B, H, 1, k)  @  (B, H, k, V) = ((B, H, k, 1) *  (B, H, k, V)).sum(-2)
50         out[:, i] = (q_t.unsqueeze(-1) * St).sum(-2)
51
52         M_prev = Mt
53         S_prev = St
54
55     o = out
56     if output_final_state:
57         final_state = torch.stack([S_prev, M_prev], dim=0)
58     else:
59         final_state = None
60
61     return o, final_state
62
```

```
63 def chunk_momentum_delta_rule(
64         q: torch.Tensor,
65         k: torch.Tensor,
66         v: torch.Tensor,
67         p: torch.Tensor,     # we use p = α · k out of this function
68         log_alpha: torch.Tensor,
69         log_mu: torch.Tensor,
70         beta: torch.Tensor,
71         eta: torch.Tensor,
72         scale: float = None,
73         initial_S: torch.Tensor = None,
74         initial_M: torch.Tensor = None,
75         output_final_state: bool = False,
76         chunk_size: int = 64,
77 ):
78     # assert not torch.any(torch.eq(beta, 0))
79     BT = chunk_size
80     if scale is None:
81         scale = 1 / (q.shape[-1] ** 0.5)
82     # Calculate padding needed to make T a multiple of BT
83     q, k, v, p, log_alpha, log_mu, beta, eta \
84         = map(lambda x: x.to(torch.float32), [q, k, v, p, log_alpha, log_mu, beta, eta])
85     T , pad_len = q.shape[1], (BT - (T % BT)) % BT
86
87     if pad_len > 0:
88         q = F.pad(q, (0, 0, 0, 0, 0, pad_len))
89         k = F.pad(k, (0, 0, 0, 0, 0, pad_len))
90         v = F.pad(v, (0, 0, 0, 0, 0, pad_len))
91         p = F.pad(p, (0, 0, 0, 0, 0, pad_len))
92         log_alpha = F.pad(log_alpha, (0, 0, 0, pad_len))
93         log_mu = F.pad(log_mu, (0, 0, 0, pad_len))
94         beta = F.pad(beta, (0, 0, 0, pad_len))
95         eta = F.pad(eta, (0, 0, 0, pad_len))
96
97     # l is the sequence lenght after padding
98     B, l, H, DK = q.shape
99     DV = v.shape[-1]
100    q = q * scale
101    assert l % chunk_size == 0
102    assert q.shape == (B, pad_len+T, H, DK)
103    assert log_alpha.shape == (B, pad_len+T, H)
104
105    k_eta = eta[..., None] * k
106    q, k, v, p, log_alpha, log_mu, beta = map(
107        lambda x: rearrange(x, 'b (n c) h d -> b h n c d', c=chunk_size),
108        [q, k_eta, v, p, log_alpha.unsqueeze(-1),
109         log_mu.unsqueeze(-1), beta.unsqueeze(-1)]
110    )
111
112    log_a_cum = log_alpha.squeeze(-1).cumsum(-1)
113    log_m_cum = log_mu.squeeze(-1).cumsum(-1)
114    log_beta  = (beta + 1e-6).squeeze(-1).log()
115
116    log_c_before = log_beta + log_m_cum - log_a_cum
117    log_ct       = torch.logcumsumexp(log_c_before, dim=-1)
118    log_ct_tm1   = torch.cat([torch.full_like(log_ct[:, :, :, :1], float('-inf')),
119                              log_ct[:, :, :, :-1]], dim=-1)
120
121    s = (log_ct_tm1.unsqueeze(-2) - log_ct.unsqueeze(-1)).tril()  # s <= 0
122    s = 1 - torch.exp(s)
123
124    log_bar_a_tm1 = torch.cat([torch.zeros_like(log_a_cum[:, :, :, :1]),
125                               log_a_cum[:, :, :, :-1]], dim=3)
126
127    b_t    = (log_a_cum + log_ct).exp()
```

```
128      b_tm1 = torch.cat([torch.zeros_like(b_t[:, :, :, :1]), b_t[:, :, :, :-1]], dim=3)
129
130      gamma_mask_q = (log_ct.unsqueeze(-1) + log_a_cum.unsqueeze(-1)
131                      - log_m_cum.unsqueeze(-2)  ).exp().float().tril() * s
132      gamma_mask   = torch.cat([torch.zeros_like(gamma_mask_q[:, :, :, :1]),
133                               gamma_mask_q[:, :, :, :-1]], dim=3)
134
135      attn = (p @ k.transpose(-1, -2)) * gamma_mask
136
137      attn_inv = -attn
138      for i in range(1, chunk_size):
139          attn_inv[..., i, :i] += (attn_inv[..., i, :, None].clone() * attn_inv[..., :, :i].
         clone()).sum(-2)
140      attn_inv = attn_inv + torch.eye(chunk_size, dtype=attn_inv.dtype, device=q.device)
141
142      alpha_tm1_p = log_bar_a_tm1.exp()[..., None] * p
143      b_tm1_p     = b_tm1[..., None] * p
144
145      u_c = attn_inv @ v
146      y_c = attn_inv @ alpha_tm1_p
147      z_c = attn_inv @ b_tm1_p
148
149      S_pre = initial_S if initial_S is not None else k.new_zeros(B, H, DK, DV)
150      M_pre = initial_M if initial_M is not None else k.new_zeros(B, H, DK, DV)
151
152      o = torch.zeros_like(v)
153      num_chunks = q.shape[2]
154      for i in range(num_chunks):
155          q_i, k_i, = q[:, :, i], k[:, :, i]
156          # Correction Value v, objective loss ||Sk - V||^2
157          v_i = u_c[:, :, i] - y_c[:, :, i] @ S_pre + z_c[:, :, i] @ M_pre
158
159          # qS read out
160          attn_inner     = (q_i @ k_i.transpose(-1, -2)) * gamma_mask_q[:, :, i]
161          bar_alpha_t_q  = q_i * log_a_cum[:, :, i, :].exp().unsqueeze(-1)
162          b_t_q          = q_i * b_t[:, :, i, :].unsqueeze(-1)
163          qS_inter       = bar_alpha_t_q @ S_pre - b_t_q @ M_pre
164          o[:, :, i]     = qS_inter + attn_inner @ v_i
165
166          # update S, M
167          decay_s = gamma_mask_q[:, :, i, -1].unsqueeze(-1)
168          S = log_a_cum[:, :, i, -1, None, None].exp() * S_pre \
169              - b_t[:, :, i, -1, None, None] * M_pre \
170              + (k_i * decay_s).transpose(-1, -2) @ v_i
171
172          decay_m = (log_m_cum[:, :, i, -1, None] - log_m_cum[:, :, i]).exp()[..., None]
173          M = log_m_cum[:, :, i, -1, None, None].exp() * M_pre \
174              - (k_i * decay_m).transpose(-1, -2) @ v_i
175
176          S_pre, M_pre = S, M
177
178      if output_final_state:
179          final_state = torch.stack([S_pre, M_pre], dim=0)
180      else:
181          final_state = None
182
183      # unpad
184      o = rearrange(o, 'b h n c d -> b (n c) h d')
185      o = o[:, :T]
186      return o, final_state
```

## F. Stability Condition of Gated Momentum Dynamics

To analyze the stability condition of the proposed stepwise momentum rule, we reformulate the coupled updates into a unified discrete state space dynamic representation. Recall the recursive updates for the momentum state $\mathbf{M}_t$ and the fast weight state $\mathbf{S}_t$:

$$\begin{aligned} \mathbf{M}_t &= \mu_t \, \mathbf{M}_{t-1} - \eta_t \, \boldsymbol{k}_t \big(\boldsymbol{v}_t - \alpha_t \mathbf{S}_{t-1}^\top \boldsymbol{k}_t\big)^\top, \\ \mathbf{S}_t &= \alpha_t \, \mathbf{S}_{t-1} - \beta_t \, \mathbf{M}_t. \end{aligned} \tag{55}$$

By substituting $\mathbf{M}_t$ into the update of $\mathbf{S}_t$, the coupled dynamics can be written explicitly as

$$\begin{aligned} \mathbf{S}_t &= \big(\alpha_t \mathbf{I} - \alpha_t \beta_t \eta_t \, \boldsymbol{k}_t \boldsymbol{k}_t^\top\big)\mathbf{S}_{t-1} - \beta_t \mu_t \, \mathbf{M}_{t-1} + \beta_t \eta_t \, \boldsymbol{k}_t \boldsymbol{v}_t^\top, \\ \mathbf{M}_t &= \mu_t \, \mathbf{M}_{t-1} + \alpha_t \eta_t \, \boldsymbol{k}_t \boldsymbol{k}_t^\top \mathbf{S}_{t-1} - \eta_t \, \boldsymbol{k}_t \boldsymbol{v}_t^\top. \end{aligned} \tag{56}$$

While the optimizer perspective is useful for motivating the recurrent structure, it offers limited guidance for the design of input-dependent gating. In our formulation, the gates $\alpha_t$ and $\beta_t$ are kept input-dependent to preserve expressivity, whereas the optimizer-related coefficients (e.g., $\mu_t$ and $\eta_t$) are treated as fixed scalars. The system can be compactly expressed in block matrix form as

$$\begin{pmatrix} \mathbf{S}_t \\ \mathbf{M}_t \end{pmatrix} = \mathbf{A} \begin{pmatrix} \mathbf{S}_{t-1} \\ \mathbf{M}_{t-1} \end{pmatrix} + \begin{pmatrix} \beta_t \eta_t \, \boldsymbol{k}_t \boldsymbol{v}_t^\top \\ -\eta_t \, \boldsymbol{k}_t \boldsymbol{v}_t^\top \end{pmatrix}, \tag{57}$$

where the state transition matrix $\mathbf{A}$ is given by

$$\mathbf{A} = \begin{pmatrix} \alpha_t \mathbf{I} - \alpha_t \beta_t \eta_t \, \boldsymbol{k}_t \boldsymbol{k}_t^\top & -\beta_t \mu_t \, \mathbf{I} \\ \alpha_t \eta_t \, \boldsymbol{k}_t \boldsymbol{k}_t^\top & \mu_t \, \mathbf{I} \end{pmatrix}. \tag{58}$$

The matrix $\mathbf{A}$ admits a closed-form spectral characterization. Its spectrum is

$$\mathrm{Spec}(\mathbf{A}) = \Big\{ \underbrace{\alpha_t}_{\times(d-1)}, \; \underbrace{\mu_t}_{\times(d-1)}, \; \lambda_+, \; \lambda_- \Big\}, \tag{59}$$

where the remaining two eigenvalues $\lambda_\pm$ are given by

$$\lambda_\pm = \frac{\alpha_t + \mu_t - \alpha_t \beta_t \eta_t \|\boldsymbol{k}_t\|^2 \pm \sqrt{\big(\alpha_t + \mu_t - \alpha_t \beta_t \eta_t \|\boldsymbol{k}_t\|^2\big)^2 - 4\alpha_t \mu_t}}{2}. \tag{60}$$

All eigenvalues satisfy $|\lambda| \leq 1$ if and only if $|\alpha_t| \leq 1$, $|\mu_t| \leq 1$, $\|\boldsymbol{k}_t\|^2 = 1$, and

$$-(1 - \alpha_t)(1 - \mu_t) \; \leq \; \alpha_t \beta_t \eta_t \|\boldsymbol{k}_t\|^2 \; \leq \; (1 + \alpha_t)(1 + \mu_t).$$

## G. Additional Experiment Details

**MQAR Experiments Details.** For the MQAR experiments, we largely follow the setup described in Arora et al. (2023a). Models are trained on sequences of 64–256 tokens containing 4–64 key–value pairs, and are evaluated on substantially more challenging settings with 512–2048 token sequences containing 32–512 key–value pairs. The hyperparameters are summarized in Table 5.

**Language Model Experiments Details.** All models are trained from scratch under two scales: 400M and 1.3B parameters, both with a sequence length of 4k. The identical training configuration is used for training all models; we use the AdamW optimizer (Loshchilov & Hutter, 2017) across all experiments. The 400M and 1.3B models are trained on 15B and 100B tokens, respectively, with batch sizes of 0.5M and 1M. The learning rate follows a cosine decay schedule, peaking at $3 \times 10^{-4}$ after a warmup phase of 0.5B tokens for the 400M model and 1B tokens for the 1.3B model and decaying to $3 \times 10^{-5}$ by the end of training. We use the GPT-2 tokenizer and train on a 100B-token subset of SlimPajama (Soboleva et al., 2023), which originally contains 627B tokens.

*Table 5.* The MQAR search hyperparameter.

| Hyper Parameter | Search |
| --- | --- |
| Embedding dimension | [128, 256] |
| Number of layers | 2 |
| Number of heads | 2 |
| Key size | 128 |
| Expand Value size | 2 |
| Epochs | 32 |
| Batch size | 256 |
| Optimizer | AdamW |
|    Learning rate | [5e-5, 1e-4, 2e-4, 5e-4, 1e-3, 2e-3, 5e-3, 1e-2, 2e-2] |
|    Weight decay | [0.1] |
|    $\beta$s | (0.9, 0.98) |
| Scheduler | Cosine Scheduler with Warmup (with default setting) |

In addition to perplexity (PPL) on WikiText (Wiki.), we evaluate models on a diverse set of downstream tasks covering commonsense reasoning and question answering, following the evaluation protocol in Yang et al. (2025). These tasks include HellaSwag (Hella.; (Zellers et al., 2019)), LAMBADA (Lamb.; (Paperno et al., 2016)), WinoGrande (Wino.; (Sakaguchi et al., 2021)), ARC-Easy (ARCe) and ARC-Challenge (ARCc; (Clark et al., 2018)), BoolQ (Clark et al., 2019) and SciQA (Auer et al., 2023).

For retrieval intensive in-context tasks, we follow the prefix-linear-attention setup of Arora et al. (2024) with 2K input tokens, and evaluate on SWDE (Lockard et al., 2019), SQuAD (Rajpurkar et al., 2016), FDA (Arora et al., 2023b), TQA (Kembhavi et al., 2017), NQ (Kwiatkowski et al., 2019), and DROP (Dua et al., 2019). We adopt the minimally transformed versions of these benchmarks from Arora et al. (2024), which are designed to support the evaluation of non instruction tuned models.

## H. Additional Experiments

Table 6 reports the complete results of the additional ablation studies on the 400M models. All experiments follow the same training setup, with only one variable modified at a time.

## I. Limitations and Future Work

While our experiments with MDN are conducted at a reasonable scale, covering both 400M and 1.3B models, we were unable to perform larger-scale experiments due to limited compute resources. It is therefore still unclear how MDN scales to larger models and datasets, especially at the 7B scale and beyond. Given that MDN retains recurrent decoding and linear-time sequence processing, we expect its efficiency–performance trade-off to remain promising at larger scales, but this needs to be verified through systematic scaling experiments.

Our current implementation also leaves room for further system optimization. MDN introduces an additional momentum state, and the present chunkwise training implementation materializes correction values to improve backward efficiency. As a result, its training throughput is still lower than highly optimized first-order linear-attention baselines such as GDN and Comba, although it remains comparable to Mamba2 and KDA. Future work will focus on more optimized kernels, memory-efficient backward strategies, and better compatibility with tensor parallelism.

In addition, our hybrid experiments only explore a small set of linear/full-attention ratios. A more complete study of layer placement, hybrid ratios, and gating parameterizations may further improve the efficiency performance trade-off. Beyond language modeling, it would also be interesting to apply MDN to other long-sequence modalities, such as speech, video, time-series, and genomics, where efficient long-range dependency modeling is important.

*Table 6.* The extended experiments of downstream tasks evaluation. The symbol "acc_n" denotes length-normalized accuracy. The commonsense reasoning tasks are performed using the LM evaluation harness (Gao et al., 2024). The in-context retrieval intensive task follows prefix-linear-attention (Arora et al., 2024) with 2K input tokens. All models are implemented and trained using the default configurations provided by the FLA (Yang & Zhang, 2024) and FLAME (Zhang & Yang, 2025) frameworks, respectively.

| Model | Perplexity | | Commonsense Reasoning Task | | | | | | | | | In-context Retrieval Task | | | | | | |
| | Lamb. ppl ↓ | Wiki. ppl ↓ | Hella. acc_n ↑ | Lamb. acc ↑ | ARCe acc ↑ | ARCc acc_n ↑ | PIQA acc ↑ | Wino. acc ↑ | BoolQ acc ↑ | SciQ acc ↑ | Avg. acc ↑ | FDA acc ↑ | SWDE acc ↑ | SQD. acc ↑ | NQ acc ↑ | TQA. acc ↑ | Drop acc ↑ | Avg. acc ↑ |
|---|---|---|---|---|---|---|---|---|---|---|---|---|---|---|---|---|---|---|
| *400M parameters model with 15B training tokens and 0.5M batch size tokens* | | | | | | | | | | | | | | | | | | |
| **Transformer** | 54.36 | 32.80 | 34.40 | 33.24 | 45.62 | 24.23 | 64.42 | 52.17 | 59.48 | 70.90 | 48.06 | 43.32 | 31.87 | 29.66 | 17.96 | 41.59 | 18.11 | 30.42 |
| **Mamba2** | 60.42 | 33.45 | 35.08 | 29.69 | 46.68 | 23.55 | 65.18 | 52.09 | 59.14 | 71.40 | 47.85 | 11.81 | 17.24 | 27.01 | 13.78 | 38.92 | 17.97 | 21.12 |
| **GDN** | 45.63 | 32.10 | 34.90 | 34.85 | 46.13 | 24.91 | 65.56 | 52.33 | 57.86 | 71.50 | 48.51 | 14.99 | 20.99 | 27.24 | 14.76 | 40.88 | 18.69 | 22.93 |
| **Comba**[1] | 46.19 | 31.73 | 35.78 | 34.31 | 47.05 | 24.66 | 65.78 | 51.54 | 58.32 | 73.80 | 48.91 | 17.08 | 20.99 | 27.18 | 16.03 | 43.78 | 19.02 | 24.01 |
| **KDA** | 43.44 | 31.96 | 35.95 | 36.62 | 47.14 | 23.89 | 65.79 | 53.28 | 56.57 | 73.20 | 49.06 | 18.44 | 23.71 | 28.12 | 15.14 | 41.35 | 20.08 | 24.47 |
| **MDN (Ours)** | 41.62 | 31.51 | 35.60 | 37.43 | 46.93 | 25.17 | 66.43 | 50.28 | 59.25 | 74.30 | **49.42** | 28.07 | 24.65 | 28.01 | 16.95 | 43.01 | 19.89 | **26.76** |
| *Ablation studies* | | | | | | | | | | | | | | | | | | |
| w/o Output Corr. | 42.31 | 31.72 | 35.41 | 36.10 | 46.42 | 24.23 | 66.81 | 51.22 | 60.76 | 72.60 | 49.19 | 23.16 | 23.24 | 27.85 | 16.38 | 43.60 | 18.88 | 25.52 |
| w/o Momen.[2] | *NaN at 1st step: parallel kernels require $\mu \neq 0$ to avoid division by zero.* | | | | | | | | | | | | | | | | | |
| w/o Momen.[3] | 47.01 | 32.11 | 35.34 | 34.97 | 47.35 | 25.09 | 65.4 | 51.78 | 59.63 | 74.50 | 49.26 | 13.44 | 16.78 | 26.27 | 10.48 | 36.08 | 17.68 | 20.12 |
| w/o clamp $\mu_{\min}^{\log}$ | *NaN at 70th step: stability issues arise without a lower bound on $\mu$.* | | | | | | | | | | | | | | | | | |
| w/o $\alpha_{\max}$ | *NaN at 1st step if fixed $\alpha_{\max} = 1$.* | | | | | | | | | | | | | | | | | |
| w/o $\beta_{\max}$ | 42.72 | 31.52 | 35.32 | 35.78 | 47.18 | 24.32 | 65.56 | 51.22 | 57.83 | 74.20 | 48.93 | 27.79 | 25.40 | 27.71 | 16.19 | 42.12 | 19.21 | 26.40 |
| $\eta_t = 2\,\mathrm{sigmoid}(\cdot)$ | 49.10 | 31.89 | 35.80 | 34.43 | 48.19 | 25.34 | 65.89 | 52.88 | 58.01 | 74.70 | 49.41 | 24.52 | 23.43 | 28.18 | 16.60 | 41.94 | 18.54 | 25.54 |
| *Sweeping the minimum clamping value of $\mu_{\min}^{\log}$* | | | | | | | | | | | | | | | | | | |
| -2 (Reported) | 41.62 | 31.51 | 35.60 | 37.43 | 46.93 | 25.17 | 66.43 | 50.28 | 59.25 | 74.30 | 49.42 | 28.07 | 24.65 | 28.01 | 16.95 | 43.01 | 19.89 | 26.76 |
| -1.5 | 42.65 | 31.50 | 35.90 | 37.07 | 46.25 | 23.89 | 64.36 | 51.54 | 59.51 | 73.00 | 48.94 | 26.07 | 23.62 | 28.85 | 17.80 | 40.76 | 19.07 | 26.03 |
| -1.357 | 44.90 | 31.44 | 35.95 | 35.44 | 47.52 | 24.40 | 65.78 | 52.17 | 60.15 | 74.60 | 49.50 | 22.80 | 25.04 | 27.38 | 15.41 | 42.42 | 18.83 | 25.31 |
| -1 | 43.53 | 31.39 | 35.53 | 35.96 | 47.22 | 25.26 | 66.27 | 52.57 | 60.89 | 75.30 | 49.88 | 20.98 | 24.18 | 28.89 | 17.74 | 43.72 | 19.59 | 25.85 |
| *Hybrid models with Linear Attention: Full Attention* | | | | | | | | | | | | | | | | | | |
| Mamba2-H (3:1) | 61.27 | 33.73 | 35.35 | 29.77 | 47.56 | 24.23 | 65.23 | 52.09 | 59.79 | 72.70 | 48.34 | 15.17 | 19.21 | 26.94 | 14.25 | 40.40 | 17.49 | 22.24 |
| GDN-H (3:1) | 46.07 | 29.96 | 35.68 | 34.74 | 46.72 | 25.17 | 65.23 | 51.22 | 57.58 | 71.90 | 48.53 | 48.68 | 38.80 | 32.68 | 18.78 | 43.66 | 17.49 | 33.35 |
| Comba-H (3:1) | 40.88 | 29.92 | 36.00 | 38.29 | 45.96 | 24.49 | 65.34 | 50.91 | 59.27 | 74.50 | 49.35 | 53.86 | 35.43 | 33.93 | 19.99 | 44.43 | 20.65 | 34.72 |
| KDA-H (3:1) | 41.78 | 29.49 | 36.00 | 37.61 | 46.42 | 24.23 | 64.91 | 52.17 | 56.61 | 73.50 | 48.93 | 52.13 | 37.77 | 33.46 | 20.56 | 44.85 | 17.97 | 34.46 |
| **MDN-H (3:1)** | 46.71 | 30.09 | 35.68 | 34.48 | 46.17 | 23.98 | 65.23 | 51.85 | 59.76 | 71.70 | 48.61 | 49.59 | 36.27 | 34.09 | 19.20 | 44.67 | 19.21 | 33.84 |
| **MDN-H (7:1)** | 42.95 | 30.32 | 36.18 | 36.86 | 47.97 | 25.60 | 65.78 | 51.38 | 58.84 | 74.80 | 49.68 | 50.95 | 38.24 | 32.75 | 18.88 | 44.31 | 21.08 | 34.37 |

1. Protocol note: Comba reports LAMBADA-Standard and WikiText-2K, while we follow GDN with LAMBADA-OpenAI and WikiText-4K (matched to training length).
2. Without momentum: directly setting $\mu = 0$.
3. Without momentum: a runnable variant obtained by setting GDN with $\alpha_{\max}$ and $\beta_{\max}$.

