# OpenReview forum: "MDN: Parallelizing Stepwise Momentum for Delta Linear Attention"
_ICML.cc/2026/Conference — ICML 2026 regular_

### Official Review · Reviewer_CzTW · 2026-03-07

**Soundness:** 3
**Presentation:** 3
**Significance:** 2
**Originality:** 3
**Overall Recommendation:** 4
**Confidence:** 4

**Summary:**

The paper introduces Momentum DeltaNet (MDN). It applies a stepwise momentum rule to the delta update in linear attention. Naive momentum prevents efficient parallelization due to nested temporal dependencies. The authors solve this by geometrically reordering the nested summations. This decouples the coefficients and enables a chunkwise parallel Triton implementation. Furthermore, the paper models the momentum update as a second-order dynamical system. The authors analyze the complex conjugate eigenvalues. They propose a quadrant constraint on the gating mechanism to ensure training stability. Experiments on 400M and 1.3B scales show improvements over GDN, Mamba2, and Comba on synthetic recall, language modeling, and long-context tasks.

**Compliance With Llm Reviewing Policy:**

Affirmed.

**Key Questions For Authors:**

1. Equation 3: The loss is defined as $\frac{1}{2}||v_t - S_{t-1}^T k_t||_2^2$. Shouldn't there be an $\alpha_t$ in the forward pass reconstruction? Eq 4 gradient uses $\alpha_t$. Please clarify this discrepancy.

2. Figure 6 shows training throughput is lower than Comba/GDN. Can you provide the exact percentage drop in TFLOPs or tokens/sec?

3. How sensitive is the model to the momentum hyperparameter $\mu_t$? You clamp the minimal value to $e^{-1}$.

4. What is the peak memory consumption of MDN compared to Mamba2 and GDN during a 4K sequence training step?

**Limitations:**

yes

**Strengths And Weaknesses:**

*   **Strengths:**

1.The math derivation is clever.

2. The paper is mostly readable. The core motivation is clear.

*   **Weaknesses:**

1. The experimental scale is limited. 1.3B parameters is slightly small for modern LLM baselines.

2.The training throughput is lower than GDN and Comba.

3. Eq (3) omits $\alpha_t$ in the reconstruction target, but Eq (4) suddenly includes it.

4. Line 194 has a typo ("derivative" instead of "derive").

5. Eq (11) introduces a mask without fully explaining its boundary conditions in the main text.

---

> ### Author Rebuttal · Authors · 2026-03-31
>
> > **W1**. The experimental scale is limited
>
> **Re**: Thank you for your constructive comments. We agree that larger-scale validation would further strengthen the paper. Our current experiments mainly follow GDN and Comba and are conducted at the same 400M and 1.3B scales. This choice ensures a fair comparison under a controlled setup, especially given limited computational resources and the need to reproduce several baselines. Due to compute constraints, we were not able to extend the study to larger models in this submission. We will release our code to support further exploration at larger scales by the community.
>
> ---
>
> > **W2** / **Q2** / **Q4**.  Comparison of Training FLOPs and Memory
>
> **Re**: Thank you for the constructive questions. Table R1 reports the detailed training throughput (tokens/s) and peak memory under the same 4K sequence training setting. Relative to GDN, MDN exhibits a 10.61% throughput drop. We attribute this gap mainly to the dual-state recurrence, which makes the corresponding backward Triton kernel more complex to implement efficiently. Importantly, we view this as an engineering-level constant factor overhead, rather than a fundamental parallel algorithmic bottleneck. Moreover, there is still room for systems level optimization for kernel implementation, including deeper fusion of backward-pass kernels, reduced intermediate state materialization, and better chunkwise reuse to lower memory traffic. We will further optimize the implementation in future work.
>
> *Table R1. Training Peak Memory and Throughput (4K Length * 4 Batch size).*
> |Model|Mem(GB)|Speed(tk/s)|Rel.(vs.GDN)
> |---|---|---|---|
> |Mamba2|37.21|36550.96|-19.64%
> |GDN|39.36|45484.07|Baseline
> |Comba|42.34|47518.05|4.47%
> |KDA|39.22|39575.41|-12.99%
> |MDN|44.89|40658.28|-10.61%
>
> ---
>
> > **W3** / **W4** / **Q1**. Typos. Eq (3) omits $α_t$, but Eq (4) suddenly includes it. Please clarify this discrepancy.
>
> **Re:** The intended objective in Eq.(3) is defined on the decayed fast weight  $\widetilde{\mathbf{S}}\_{t-1} = α\_t \mathbf{S}\_{t-1}$ (Line 189), rather than directly on $\mathbf{S}\_{t-1}$. In other words, the Eq.(3) should be written as $\mathcal{L}(\widetilde{\mathbf{S}}\_{t-1} ) = \frac{1}{2} \| \boldsymbol{v}\_t -\widetilde{\mathbf{S}}\_{t-1}^{\top} \boldsymbol{k}\_t \|\_2^2$ with $\widetilde{\mathbf{S}}\_{t-1} = α_t \mathbf{S}\_{t-1}$.
>
> Therefore, for decayed fast weight, the gradients are  $\nabla\_{\widetilde{\mathbf{S}}} \mathcal{L}(\widetilde{\mathbf{S}}\_{t-1} ) =  - \boldsymbol{k}\_t (\boldsymbol{v}\_t - \widetilde{\mathbf{S}}\_{t-1}^{\top} \boldsymbol{k}\_t)^{\top} = - \boldsymbol{k}\_t (\boldsymbol{v}\_t - α\_t \mathbf{S}\_{t-1}^{\top} \boldsymbol{k}\_t)^{\top}$, which is exactly the term used in Eq (4). We will revise the presentation to make the derivation clearer and easier to follow. Furthermore, we will fix the typo on Line 194 and perform a broader proofreading pass over the manuscript.
>
> ---
>
> > **W5**. Boundary conditions on Eq (11)
>
> **Re**: Thanks for your careful suggestion. We agree that the main-text explanation around Eq (11) is too brief. The key point is that the mask $\Gamma_{[t]}$ in Eq (11) is lower triangular, and the values outside the causal region are zero. Its entries are inherited from the element-wise coefficients $\gamma_{t,i}$ in Eq. (9), so Eq (11) is simply the chunkwise matrix form of the same causal boundary condition. While the full derivation is provided in Appendix D (Eqs. 53–54), we will revise the discussion around Eq (11) to state this boundary behavior explicitly, and make the transition from the element-wise definition to the chunkwise implementation easier to follow.
>
> ---
>
> > **Q3**. Sensitivity on $\log\mu$
>
> **Re**: Thank you for this constructive question. To directly assess sensitivity, we conducted an ablation study on the minimum $\log \mu$ clamp using our 400M model (Table R2).
>
> 1. Empirical Robustness: The model is remarkably stable with respect to this hyperparameter. Across the clamp range of -2 to -1, average performance varies by less than 0.5%. While a stricter clamp (-1) slightly improves certain standard language modeling tasks, our reported setting of -2 consistently provides the best overall tradeoff across varying sequence lengths.
>
> 2. Mathematical Necessity: The clamp ensures numerical robustness. Eq (9)' s log-domain prefix sums would trigger FP32 underflow as $\log \mu \to -\infty$. Small $\mu$ also weakens momentum and destabilizes backward passes. Thus, the clamp is a required parallelization guardrail, not a fragile hyperparameter.
>
> *Table R2. Sensitivity to the $\log⁡ μ$ minimum clamp value. (Full table in Link)*
> |minimum$\log μ$|Lamb.ppl↓|Wiki.ppl↓|LMAvg.acc↑|RetrievalAvg.acc↑
> |---|---|---|---|---
> |-2(reported)|**41.62**|31.51|49.42|**26.76**
> |-1.5|42.65|31.50|48.94|26.03
> |-1.357|44.90|31.44|49.50|25.31
> |-1|43.53|**31.39**|**49.88**|25.85
>
> ---
>
> [Anonymous Link] https://anonymous.4open.science/r/MDN_rebuttal-734E

---

> > ### Author Rebuttal · Reviewer_CzTW · 2026-04-04
> >
> > Thank you for the detailed response. All of my concerns have been addressed. I maintain my score recommending weak acceptance.

---

### Official Review · Reviewer_BkN2 · 2026-03-12

**Soundness:** 4
**Presentation:** 3
**Significance:** 3
**Originality:** 3
**Overall Recommendation:** 5
**Confidence:** 4

**Summary:**

This paper proposes Momentum DeltaNet (MDN), a new linear attention architecture that extends recent Delta-style recurrent models with second-order momentum dynamics. Building on the view that these architectures implicitly perform an optimization-like update, the paper shows how momentum can be introduced into the recurrence and derives a chunkwise parallel algorithm for efficient training. The authors evaluate MDN at 400M and 1.3B scale against softmax attention and several linear-attention baselines on language modeling, commonsense reasoning, in-context retrieval, and long-context benchmarks. Empirically, MDN achieves consistently strong results and outperforms most linear baselines on most tasks, while retaining competitive training and decoding efficiency.

**Compliance With Llm Reviewing Policy:**

Affirmed.

**Final Justification:**

The authors addressed my concerns, and I remain positive to the paper.

**Key Questions For Authors:**

1. Can the authors provide a stronger ablation that isolates momentum itself? For example, a comparison between first-order and second-order variants under the same architecture and parameter budget, rather than only removing output correction or noting that unconstrained gating diverges.

2. Would the model benefit from other type of constraints (e.g. explicitly caping the trajectories on the right half-plane, or loosening the quadrant constraints to make the trajectories span a wider region in the right half-plan)? A more detailed discussion on this would be appreciated.

**Limitations:**

Yes

**Strengths And Weaknesses:**

Strengths

1. The paper is technically solid overall, combining a nontrivial derivation of momentum-based recurrent updates and chunkwise parallel training with broad empirical validation across modeling, retrieval, long-context, efficiency, and ablation experiments.

2. The paper is generally well structured and communicates its main idea clearly, especially the motivation for stepwise momentum and its distinction from prior blockwise approaches, though the exposition could still be polished further.

3. The work addresses an important problem in efficient long-context sequence modeling by proposing a way to incorporate momentum into linear/recurrent attention without sacrificing causal consistency or scalable training.

4. The combination of stepwise momentum in Delta-style linear attention, exact chunkwise parallelization, and a second-order dynamical interpretation makes this feel like a genuinely new architectural contribution rather than a minor variant.

Weaknesses

1. The authors reported in section 3.2 that the unconstrained momentum system results in catastrophic numerical failures during training. From Figure 2, if only the 2nd and 3rd quadrants are damaging performance, the middle figure shows that only a small portion of the roots are in the left half-plane, whereas the right figure shows that the trajectories are overly crowded in the right half-plane. However, little ablation is made over the constraints applied.

2. The paper only presented results against pure linear attention models, whereas KDA, gated DeltaNet and Mamba2 are all building blocks of hybrid models. It would make the contribution more solid if a hybrid variant of MDN is compared against those methods.

---

> ### Author Rebuttal · Authors · 2026-03-31
>
> > **W1**/**Q1** However, little ablation is made over the constraints applied. Can the authors provide a stronger ablation that isolates momentum itself?
>
> **Re**: Thank you for this helpful suggestion. To address this, we performed a more targeted ablation by varying the momentum-related components ($μ$, $η$, $α_\max$, $β_\max$) while keeping the overall architecture and parameter budget fixed.
>
> As shown in Table R1, removing or weakening these components leads either to training divergence or to consistent performance degradation. In particular, the constraints on $α_\max$ and $β_\max$ are important crucial for stabilizing the fast-weight dynamics: removing constraints on $α_\max$ causes immediate divergence, while removing constraints on $β_\max$ degrades performance. This stability is also supported by the fast-weight norm plots provided in the anonymous link.
>
> Furthermore, to isolate the role of momentum as directly as possible in a runnable setting, we also include the "GDN + $α_\max$ & $β_\max$" variant, which can be viewed as the closest non-momentum counterpart under the same overall architecture and parameter budget. Compared with this variant, full MDN achieves stronger overall results, suggesting that the gain is not explained by the stabilizing constraints alone, but depends fundamentally on the momentum mechanism itself.
>
> Taken together, these ablations suggest that the gain is not explained by output correction or gating constraints alone, but depends on the momentum mechanism together with its stability aware parameterization. We will include the full ablation table and discuss this more explicitly in the revised manuscript.
>
> *Table R1. Ablation of the momentum-related components. (Full Table in Link)*
> | Model | Lamb. ppl ↓ | Wiki. ppl ↓ | LM Avg. acc ↑ | Retrieval Avg. acc ↑ |
> |---|---|---|---|---|
> |MDN 400M model|**41.62**|**31.51** | **49.42** | **26.76** |
> | w/o Output Correc. | 42.31 | 31.72 | 49.19 | 25.52 |
> | w/o Momentum (Fixed $μ=0$) | NaN at 1st step | - | - | - |
> | w/o Momentum (GDN + $α_\max$ & $β_\max$) | 47.01 | 32.11 | 49.26 | 20.12 |
> | w/o clamp $μ$ | NaN at 70th step | - | - | - |
> | w/o $α_\max$ (Fixed $α_\max=1$) | NaN at 1st step| - | - | - |
> | w/o $β_\max$ (Fixed $β_\max=1$) | 42.72 | 31.52 | 48.93 | 26.40 |
> | $η = tanh(·)+1 \rightarrow η = 2·sigmoid(·) $| 49.10 | 31.89 | 49.41 | 25.54 |
>
> ---
>
> > **W2** It would make the contribution more solid if a hybrid variant of MDN is compared against those methods.
>
> **Re:** Thank you for this valuable suggestion. To address this point, we conducted additional 400M hybrid-model experiments by combining MDN with full attention (Table R2). Following prior hybrid designs such as Kimi Linear (KDA + full attention) and Qwen3.5 (GDN + full attention), we first adopted the same 3:1 linear/full ratio as a standard baseline. Under this setting, MDN + Full (3:1) remains competitive with strong hybrid baselines. When increasing the ratio to 7:1 further improves MDN to the best LM Avg, while maintaining competitive retrieval performance. This suggests that MDN itself already provides stronger modeling capacity, so a high density of full attention may be unnecessary.
>
> Overall, these results indicate that MDN is not limited to pure linear-attention settings, but can also serve as a strong building block in hybrid architectures. We believe that more fully exploring the hybrid design space for MDN is an important direction for future work.
>
> *Table R2. Additional 400M hybrid-model with 3:1 ratio results. (Full table on link).*
> |Model|Lamb. ppl ↓| Wiki. ppl ↓ | LM Avg. acc ↑ | Retrival Avg. acc ↑ |
> |---|---|---|---|---|
> |Trans++|54.36|32.80 | 48.06 | 30.42 |
> |Mamba2-H| 61.27 | 33.73 | 48.34 | 22.24 |
> |GDN-H | 46.07 | 29.96 | 48.53 | 33.35 |
> |Comba-H| **40.88** | 29.92 | 49.35 | **34.72**|
> |KDA-H|41.78 | **29.49** | 48.93 | 34.46 |
> |MDN-H|46.71|30.09|48.61|33.84|
> |MDN-H (7:1) |42.95| 30.32 | **49.68** | 34.37|
>
> ---
>
> **Q2**. Would the model benefit from other type of constraints?
>
> **Re:** We agree that other constraint designs may also be beneficial. Our current constraint is intended as a conservative but practical stability-preserving parameterization, rather than a unique solution. For example, explicitly constraining the trajectories to stay in the right half-plane by ensuring that the eigenvalues of the transition matrix have positive real parts, or relaxing the current quadrant constraint to cover a wider stable region, could potentially improve expressivity. However, such less conservative designs may also introduce stronger transient amplification and optimization instability, and would require additional analysis to make them practical for large-scale training. Since the main focus of this work is to address the parallelization challenge of stepwise momentum, we leave the exploration of alternative constraint designs as an important direction for future work.
>
> ---
>
> [Anonymous Link] https://anonymous.4open.science/r/MDN_rebuttal-734E

---

> > ### Author Rebuttal · Reviewer_BkN2 · 2026-04-04
> >
> > The authors provided additional ablations that better isolate the impact of momentum and clarify the impact of the stability constraints. They also included new hybrid-model results showing MDN works well in that setting. These address my main concerns.

---

### Official Review · Reviewer_bW3P · 2026-03-12

**Soundness:** 4
**Presentation:** 3
**Significance:** 3
**Originality:** 2
**Overall Recommendation:** 5
**Confidence:** 4

**Summary:**

This paper proposes a novel linear recurrent architecture called Momentum DeltaNet which takes inspiration from the test-time training perspective on the DeltaNet recurrence and applies stepwise momentum to it which induces a second-order linear recurrence. The authors derive an efficient chunkwise parallel form for computing this recurrence by reordering coefficients in the recurrence. Furthermore, they propose a stable parameterization of it by analyzing the recurrent eigenvalues. Several experiments are ran to demonstrate the empirical strength of this linear recurrence against existing ones.

**Compliance With Llm Reviewing Policy:**

Affirmed.

**Final Justification:**

The only reservation I had was lack of implementation and they gave it to me so this is a clear accept in my opinion. The rest of my thoughts have been given in my response to their review.

**Key Questions For Authors:**

1. In Figure 1, are the authors making two distinct claims? One that the non-linear recurrences in the TTT literature needs to make the minibatch approximation to be chunkwise parallel and two that there is a gap (i.e. mathematically different in infinite precision) between the chunkwise parallel form of the recurrence used during training and the fully recurrent mode used during inference? Please clarify. And given that the authors are bringing up some of these TTT models (e.g. LaCT, LMM, Titans, Atlas), can they address why they don't need to be included in the baseline set? I agree that they don't, but some clarification here is necessary.
2. Can the authors provide further intuition and optimally run some ablations to provide intuition as to how they converged upon the preise functional form of the gates? Also, can the authors make some plot of either the norm of $S_t$ as a function of time or the recurrent matrix singular values as a function of time to demonstrate stability? Eigenvalues are only sufficient in the case of symmetric matrices.
3. Can the authors provide a full table of evals for the ablations on the output correction factor from Comba [1]? From the results it seems that this is resulting in the largest performance improvement, yet there is only an aggregated table in Table 4
4. Where is the ablation that shows "In contrast, removing the gating constraint leads to training divergence, confirming its critical role in stabilizing." ?
5. Could the authors comment on protocol differences that may explain why the WikiText and LAMBADA perplexities in Table 2 differ from those reported in the Comba paper? Since Table 2 states that all models are retrained under shared FLA/FLAME defaults and equal token budgets, some clarification on this discrepancy would be helpful.

References:

[1] Comba: Improving Bilinear RNNs with Closed-loop Control

**Limitations:**

yes

**Strengths And Weaknesses:**

Strengths:
1. This paper leans on the literature from test-time training/regression and applies a clean, yet simple idea from optimization by using momentum. It is a very natural extension of previous work
2. This paper finds a nice trick to simplify the derivation of the chunkwise parallel form to avoid working with 2x2 prefix products and applying WY to that (although I believe this messier approach should also be doable)
3. The empirical results are thorough and look to be solid across the board
4. The paper is quite easy to follow and follows the format of previous papers in the space which is good

Weaknesses:
1. The main weakness is the lack of code. I am assuming the authors integrated this implementation with the flash-linear-attention (FLA) repository and thus would highly benefit from releasing at least the kernels for the recurrence and layer implementation (and optimally the FLAME training configs they used as well) given that the core contribution of this work is the efficient implementation of the recurrence in Triton and integration into FLA. Without code, it is quite non-trivial to reproduce the results in the paper and/or use the recurrence for further iteration.
2. Figure 1 is quite confusing, it's unclear what the authors are trying to convey with this. See questions below
3. The stability analysis is interesting, but not fully convincing as presented. See questions below for more
4. There are no ablations presented on the parameterization of the functional form of the "Stability Aware Gating Parameterization" presented in Section 3.3. See questions below for more
5. The paper would benefit from a stronger round of editing and mathematical cleanup.

---

> ### Author Rebuttal · Authors · 2026-03-31
>
> > **W1**. The main weakness is the lack of code.
>
> **Re**: We agree that code release is important for reproducibility, especially because the Triton implementation is a central part of the contribution. We plan to release the code upon acceptance, including the recurrence kernel, the layer implementation, and the training configurations used in our experiments.
>
> ---
>
> > **W2**/**Q1**. In Figure 1, are the authors making two distinct claims? Can they address why they don't need to be included in the baseline set?
>
> **Re**: Thank you for this insightful question. Yes, indeed, Figure 1 is meant to make two related points: (1) some Typical TTT / nonlinear recurrent methods rely on blockwise or sliding-window approximations for scalable parallel training, and (2) these approximations may create a gap between training-time parallel updates and strict recurrent decoding. Our point is not that these methods are invalid, but that there is a trade-off between efficiency and training–inference consistency, and MDN is designed to preserve both via an exact chunkwise parallel formulation of the strict stepwise recurrence.
>
> For baselines, we focus on directly comparable linear-attention architectures under the same training setup. The TTT-style models in Figure 1 are included for conceptual positioning, rather than as the primary target class of our experimental comparison. We will revise Figure 1 and its caption to make this distinction clearer.
>
> ---
>
> > **W3**/**W4**/**Q2**/**Q4**. Ablation on the functional form of the gates?  Can the authors make a plot of either the norm/singular values as a function of time to demonstrate stability?
>
> **Re**: Thank you for this helpful suggestion. To address this, we performed a more targeted ablation on the key components of the stability-aware gating parameterization ($μ$, $η$, $α_\max$, $β_\max$) while keeping the overall architecture and parameter budget fixed.
>
> As shown in Table R1 on Reviewer BkN2 (**Full Table in Anonymous Link**), removing or weakening these components leads either to training divergence or to consistent performance degradation. In particular, the constraints on $α_\max$ and $β_\max$ are important for stabilizing the fast-weight dynamics: removing $α_\max$ causes immediate divergence, while removing $β_\max$ degrades performance.
>
> To further illustrate stability, we also provide the state norm and largest singular value plots in Figures R1 and R2 in **Anonymous Link**. Without the $α_\max$/$β_\max$ stability constraint, both the norm and the largest singular value grow rapidly over time. With the constraint, they remain bounded. This provides additional empirical evidence that the gating constraint is critical for stable dynamics beyond the eigenvalue analysis alone. In addition, removing the lower bound on μ also makes training unstable in practice, which is consistent with the spectral analysis in Section 3.2.
>
> Thanks again for your constructive comments and insightfull feedback, which have greatly improved the quality of our manuscripts. We will incorporate these ablation analyses and plots more explicitly into the revised manuscript.
>
> ---
>
> > **Q3**. Can the authors provide a full table of evals for the ablations on the output correction factor from Comba?
>
> **Re**: Thank you for this valuable suggestion. We agree that the current aggregated presentation in Table 4 is too compressed. Due to the rebuttal space limit, we provide the full evaluation table at the anonymous link. As shown there, MDN without output correction remains competitive with Comba, indicating that the improvement is not primarily driven by output correction alone. Instead, the results suggest that the stepwise momentum rule contributes independently and remains complementary to output correction. We will make this distinction clearer in the revision and include a more comprehensive evaluation table for the output-correction ablation, together with the additional ablation experiments.
>
> ---
>
> > **Q5**. Could the authors comment on protocol differences?
>
> **Re**: Thank you for this suggestion. The discrepancy mainly comes from evaluation protocol differences rather than the shared retraining setup itself. For LAMBADA perplexity, Comba reports results on LAMBADA-standard (as noted in Table 6 of the Comba paper), while we follow GDN and evaluate on LAMBADA-openai. For WikiText perplexity, Comba uses a 2K input length, whereas we use 4K to keep the evaluation context length consistent with training. We will clarify these protocol differences explicitly in the revised manuscript.
>
> ---
>
> > **W5**. The paper would benefit from a stronger round of editing and mathematical cleanup.
>
> **Re**: Thank you for your constructive suggestion. We will carefully revise the paper for clarity, notation consistency, and mathematical presentation.
>
> ---
>
> [Anonymous Link] https://anonymous.4open.science/r/MDN_rebuttal-734E

---

> > ### Author Rebuttal · Reviewer_bW3P · 2026-03-31
> >
> > Thank you for the detailed response and the additional plotting and figures. All of my concerns have been addressed outside of the code availability. I maintain my score recommending weak acceptance solely due to lack of current reproducibility.

---

> > > ### Author Response · Authors · 2026-04-05
> > >
> > > We sincerely thank you for your careful review. We fully understand your concern regarding code availability for reproduction, and we take reproducibility seriously. While our standard practice is to open-source the code upon publication, we have expedited the release to facilitate your review. We have prepared a fully functional, anonymized repository here: https://anonymous.4open.science/r/MDN-Code
> > >
> > > To ensure straightforward reproduction, this preliminary anonymized release includes: (1) PyTorch implementations: both chunkwise and recurrent forms. (2) Triton Kernels: The optimized Triton implementation. (3) Core Architecture: The full layer implementation with the exact gate constraints. (4) Reproduction Pipeline: The complete training pipeline and all hyperparameter configurations. We will migrate this to a public GitHub repository upon acceptance.
> > >
> > > Since you noted that the lack of code was the sole factor keeping your recommendation at a weak accept, we hope this immediate release fully resolves that final reservation. If you feel that this concern has now been sufficiently addressed, we would greatly appreciate your reconsideration.
> > >
> > > Thank you again for your time and for helping us improve the manuscript.

---

### Official Review · Reviewer_LfLT · 2026-03-13

**Soundness:** 3
**Presentation:** 3
**Significance:** 3
**Originality:** 3
**Overall Recommendation:** 4
**Confidence:** 4

**Summary:**

Addressing the core pain points of existing linear attention mechanism models, this paper proposes the Momentum DeltaNet (MDN) model with three key contributions: first, designing a chunkwise parallel algorithm that geometrically rearranges update coefficients, which strictly preserves the causality of stepwise momentum while breaking through the efficiency bottleneck of sequential updates to support large-scale training; second, from the perspective of second-order dynamical systems, revealing the role of complex conjugate eigenvalues (introduced by momentum) in expanding the model's expressive capacity, and proposing quadrant gating constraints to ensure training stability; third, achieving efficient engineering deployment based on Triton kernels. Under 400M and 1.3B parameter scales, MDN consistently outperforms strong baselines such as Transformers, Mamba2, and GDN in tasks including language modeling, commonsense reasoning, and long-context retrieval, while achieving training throughput comparable to mainstream LA models, striking a balance between performance and efficiency.

**Compliance With Llm Reviewing Policy:**

Affirmed.

**Final Justification:**

Given the review of the paper and the author's response to my core concerns, I believe this paper can be ‘weak accept’.

**Key Questions For Authors:**

1.Will the efficiency disadvantage caused by dual-state computation be amplified under ultra-large parameter scales (7B/13B)? If so, are there clear optimization routes?

2.What is the generalization performance of the gating constraints? Can supplementary theoretical derivations for generalization or adaptive adjustment schemes be provided?

3.How does MDN perform on multilingual/domain-specific datasets and hybrid attention models? Can it maintain performance competitiveness while retaining efficiency advantages?

4.What is the reason for not adopting traditional optimizers such as Nesterov momentum and Adam—are there technical barriers or no performance gains verified through empirical testing?

**Limitations:**

NO

1. The efficiency impact of dual-state computation in ultra-large-scale scenarios and feasible mitigation directions;

2. The applicable boundaries of gating constraints and response strategies for failure scenarios;

3. Verification of the method's adaptability in multilingual/domain-specific scenarios.

**Strengths And Weaknesses:**

Strengths

Soundness: Supported by the Test-Time Training (TTT) perspective, it resolves the contradiction between parallelization and causality through techniques such as geometric rearrangement and log-domain stabilization. The derivation process is complete, with detailed proofs and pseudocode provided in the appendix; the second-order dynamical system analysis is reasonable, clarifying the modeling value of complex conjugate eigenvalues for the first time. The gating constraints are derived based on eigenvalue trajectories, effectively avoiding training divergence.

Presentation: Core concepts are explained in an accessible manner, with tables and figures aiding understanding, significantly lowering the theoretical barrier.

Significance: It directly targets the core bottleneck in the LA field; the proposed parallel algorithm can be migrated to other recurrent LA models, demonstrating strong generalizability; the second-order dynamical system analysis provides a new theoretical perspective, advancing the theoretical development of the field.

Originality: It achieves the parallel integration of stepwise momentum and Delta attention for the first time, balancing causality and training efficiency with breakthrough value; the second-order dynamical system perspective fills the gap in the dynamical analysis of LA models.

Weaknesses

Soundness: It has not verified efficiency changes under ultra-large parameter scales (7B/13B); the generalization of gating constraints is insufficiently demonstrated, with no analysis of the optimal value range across different tasks and data distributions; the experimental coverage is limited, as training is only conducted on the SlimPajama dataset without testing on multilingual/domain-specific data or comparing with hybrid attention models.

Presentation: Some theoretical derivations are abrupt—for example, the key steps of geometric rearrangement lack detailed explanations of geometric significance and mathematical equivalence; critical experimental hyperparameters are not fully disclosed, which may affect result reproducibility; discussions on limitations are brief, with no systematic analysis of the inherent flaws of the method and the difficulty of extending to multiple scenarios.

Significance: It has not explored the adaptability of advanced optimizers such as Nesterov momentum and Adam, limiting its generalizability; the upper limit of long-sequence experiments is only 16K, with no verification of model performance under extremely long sequences (32K/64K).

Originality: The core technology is a creative combination of existing modules, lacking completely original technical components;

---

> ### Author Rebuttal · Authors · 2026-03-31
>
> > **W1**/**Q1**. The efficiency under ultra-large parameter scales.
>
> **Re**: We additionally benchmarked a 7B model (32 layers, 4096 hidden size, 32 heads with 128 head dimension) on a single H100 GPU (4K sequence length, batch size 1).  As shown in Table R1, there is no efficiency disadvantage when scaling up to a larger model, MDN reaches comparable training throughput. This indicates that the dual-state design adds a manageable constant-factor overhead, not a fundamental scaling bottleneck. Further kernel efficiency gains are likely through backward-kernel fusion, reduced state materialization, and better chunkwise reuse.
>
> *Table R1. Training throughput on 7B model.*
> |Model|Memory(GB)|Throughput(tokens/s)|
> |---|---|---|
> |GDN|38.22|1532.1|
> |KDA|40.85|1264.9|
> |MDN|41.37|1409.8|
>
> ---
>
> > **W1**/**Q2**. The generalization of gating constraints.
>
> **Re**: Our constraint is a stability-preserving feasible region, not a dataset-specific hyperparameter to be retuned per task. Its motivation is theoretical: spectral analysis of the second-order dynamics shows that unconstrained coefficients can move eigenvalues into the left half-plane, causing sign-flipping and instability. We do not retune the constraint across benchmarks, and a sensitivity study on the minimum logμ clamp (Table R2 in the response to Reviewer CzTW; full table at the anonymous link) shows no catastrophic degradation within a reasonable range. Adaptive tuning within this stable region remains future work.
>
> ---
>
> > **W1**/**Q3**.  Multilingual/domain-specific and hybrid model
>
> **Re**: For domain-specific evaluation, we include code tasks from LongBench, where MDN achieves the best performance on both LCC and RBP (Table 3). For hybrid architectures, we additionally combine MDN with full attention at 400M scale (Table R2 in the response to Reviewer BkN2; Full table in Link). Using the same 3:1 linear/full ratio as prior hybrid designs such as Kimi Linear and Qwen, MDN + Full remains competitive; increasing the ratio to 7:1 further improves LM Avg while maintaining competitive retrieval performance. This suggests that MDN can remain effective in hybrid settings with less reliance on dense full attention. Multilingual evaluation is currently missing. We leave this to future work with a matched multilingual tokenizer and pretraining setup.
>
> ---
>
> > **W2**. Geometric significance and mathematical equivalence
>
> **Re**: For geometric perspective, the key point is that $\sum_{i=1}^{t} \sum_{j=1}^{i} a_i \cdot b_j$ accumulates over the same lower-triangular region in the (i, j) index plane. The key reordering changes the traversal from row-wise accumulation traveling (fixing i and summing over j) to column-wise accumulation traveling over (fixing j and summing over i) over the same lower-triangular region: $\sum_{i=1}^{t} \sum_{j=1}^{i} a_i \cdot b_j = \sum_{j=1}^{t} \sum_{i=j}^{t} a_i \cdot b_j = \sum_{i=1}^{t} \sum_{j=i}^{t} a_j \cdot b_i $. Thus, after renaming the index, the reordering preserves mathematical equivalence and enables further exact chunkwise parallelization. We will make this intuition more explicit in the revision.
>
> ---
>
> > **W2**. Hyperparameters and Limitations
>
> **Re**: The manuscript includes a summary of the main training details in Appendix G. We will release the full training configurations and pipeline together with the codebase. We also agree that the discussion of limitations is too brief. We will make more explicit in the revision.
>
> ---
>
> > **W3**/**Q4**. Adopting advanced optimizers.
>
> **Re**: We strongly agree that advanced optimizers like Adam or Muon would theoretically yield superior optimization trajectories. It is easy to formulate a recurrent update for linear attention based on a more advanced optimizer. However, the parallel algorithmis the key bottleneck for training more complex RNNs. We believe that solving these parallelization challenges is a highly promising direction for future work in the community.
>
> ---
>
> > **W3**. Lack extremely long sequences (32K/64K).
>
> **Re**: We additionally evaluate MDN on RULER at 32K and 64K context lengths. As shown in the Table on Anonymous Link, MDN remains competitive overall at these longer context lengths and achieves the best results on several tasks.
>
> ---
>
> > **W4**. Lacking completely original technical components
>
> **Re**: We respectfully clarify that the contribution is not a simple combination of existing modules. The novelty lies in making stepwise momentum work in delta-based linear attention through (1) a new exact chunkwise-parallel formulation based on geometric rearrangement, (2) a second-order dynamical formulation that differs qualitatively from prior first-order LA models, and (3) a stability-aware gating design that makes this higher-order recurrence train robust at scale. In this sense, the contribution is algorithmic and architectural, rather than a mere composition of known components.
>
> ---
>
> [Anonymous Link] https://anonymous.4open.science/r/MDN_rebuttal-734E

---

> > ### Author Rebuttal · Reviewer_LfLT · 2026-04-05
> >
> > The authors have thoroughly addressed my core concerns, and I will maintain my score.

---

### Decision · Program_Chairs · 2026-04-30

**Decision:**

Accept (regular)

**Comment:**

This work presents MDN, an advanced linear attention framework that enriches recent Delta-style recurrent models by incorporating second-order momentum dynamics. By framing recurrent operations as implicit optimization steps, the authors elegantly connect stepwise momentum into the model's core operation. To bridge the gap between theoretical formulation and practical scalability, the paper further derives chunkwise parallel algorithm that facilitates highly efficient training.

The reviewers identified several areas for future improvement, such as validation beyond the 1.3B parameter scale and comparisons with hybrid attention baselines. Nevertheless, reviewers unanimously reached a consensus on the paper's core innovations and practical utility. Reflecting the strong technical merits and the original nature of the proposed method, the final recommendation is an Accept. The authors are strongly encouraged to release their source code and address the minor mathematical typos in the camera-ready version.